# The *Drosophila* proventriculus lacks stem cells but compensates for age-related cell loss via endoreplication-mediated cell growth

Ben Ewen-Campen[1] ✉, Weihang Chen[1], Sudhir Gopal Tattikota[1], Ying Liu [1],
Yanhui Hu [1] & Norbert Perrimon [1,2] ✉

The *Drosophila* proventriculus is a bulb-shaped structure at the juncture of the foregut and the midgut, which plays important roles in ingestion, peritrophic membrane synthesis, and the immune response to oral pathogens. A previous study identified a population of cells in the proventriculus which incorporate bromodeoxyuridine (BrdU), a marker of DNA synthesis, and proposed that these cycling cells are multipotent stem cells that replace dying cells elsewhere in the tissue. Here, we re-investigate these cycling cells and find that they do not undergo mitosis, do not generate clonal lineages, and do not proliferate in response to tissue damage, and are therefore not stem cells. Instead, we find that these cells continually endocycle throughout the fly's life, increasing their ploidy and size, while at the same time cells in this tissue are lost into the gut lumen as the fly ages. Functionally, these cells play a critical role in the synthesis of peritrophic membrane components, and we show that when their endocycling is experimentally increased or decreased, there is a concomitant change in ploidy, tissue size, and peritrophic membrane synthesis. Further, we show that inhibition of endocycling makes flies more susceptible to orally infectious bacteria. Altogether, we show that continual endocycling of these cells is critical for maintaining tissue size and function in the face of cell loss due to aging or tissue damage.

The *Drosophila* gut is a highly regionalized organ with distinct functional domains along its length[1–3]. Within each domain, in addition to primary roles such as nutrient absorption and immunity, there are cellular mechanisms to ensure that the gut is repaired and renewed following tissue damage or cell loss. In the midgut, there are multipotent intestinal stem cells (ISCs) that continually divide to replace the differentiated cell types of the gut, both during homeostasis[4,5] and in response to tissue damage[6,7]. Analogous ISCs are present in the

mammalian small intestine[8,9], which is functionally comparable to the *Drosophila* midgut[10].

In contrast to the midgut, in the hindgut, there is no evidence of multipotent stem cells[11,12]. Instead, hindgut cells respond to tissue damage by undergoing extensive endocycling, a specialized cell cycle in which DNA is replicated without undergoing mitosis, thus increasing the ploidy and nuclear size of hindgut cells[11]. In *Drosophila*, this endocycling response to tissue damage or wounding is conserved in

[1]Department of Genetics, Blavatnik Institute, Harvard Medical School, Boston, MA, USA. [2]Howard Hughes Medical Institute, Boston, MA, USA.
✉e-mail: bewencampen@genetics.med.harvard.edu; perrimon@genetics.med.harvard.edu

the adult epidermis[11,13,14], and is important to maintain tissue integrity in the absence of stem cell activity.

At the anterior of the gut, where the foregut meets the midgut, there is a specialized bulb-shaped structure called the proventriculus (also known as the cardia, based on its shape). A previous study[15] identified a ring of cells in the proventriculus that incorporates the nucleotide analog BrdU, a marker of S phase, and proposed that these cycling cells are stem cells that replace dying cells elsewhere in the proventriculus.

Here, we re-investigate the function of these previously identified cycling cells in the proventriculus. We find no evidence that these cells undergo mitosis, generate clonal lineages, or proliferate in response to tissue damage in other regions of the proventriculus. Instead, we find that these cells continually endocycle throughout the fly's life, at a rate that varies with systemic nutritional signals. Further, we find that cells from this region of the proventriculus are progressively lost into the gut lumen as the fly ages, and that this cell loss drives nearby cells to

endocycle and thereby increase in ploidy and size. Lastly, we show that blocking endocycling in these cells reduces peritrophic membrane synthesis, whereas inappropriately enhancing endocycling leads to increased peritrophic membrane synthesis. We conclude that the proventriculus does not contain stem cells, but that it instead compensates for age-related cell loss by endocycling-mediated cell growth to maintain tissue function.

## Results and discussion
### A ring of cycling cells in Zone 4
The proventriculus consists of a doubly-folded tubular epithelium and associated muscles and neuronal structures (Fig. 1A)[16–18]. It forms during embryogenesis by the invagination of an anterior, ectodermal portion of the embryonic gut (the future esophagus) into an adjacent segment of endodermal gut, creating a bulb structure with an internal lumen[19–21]. In addition to serving as a muscular valve surrounding the esophagus, the proventriculus plays a critical role in the synthesis of

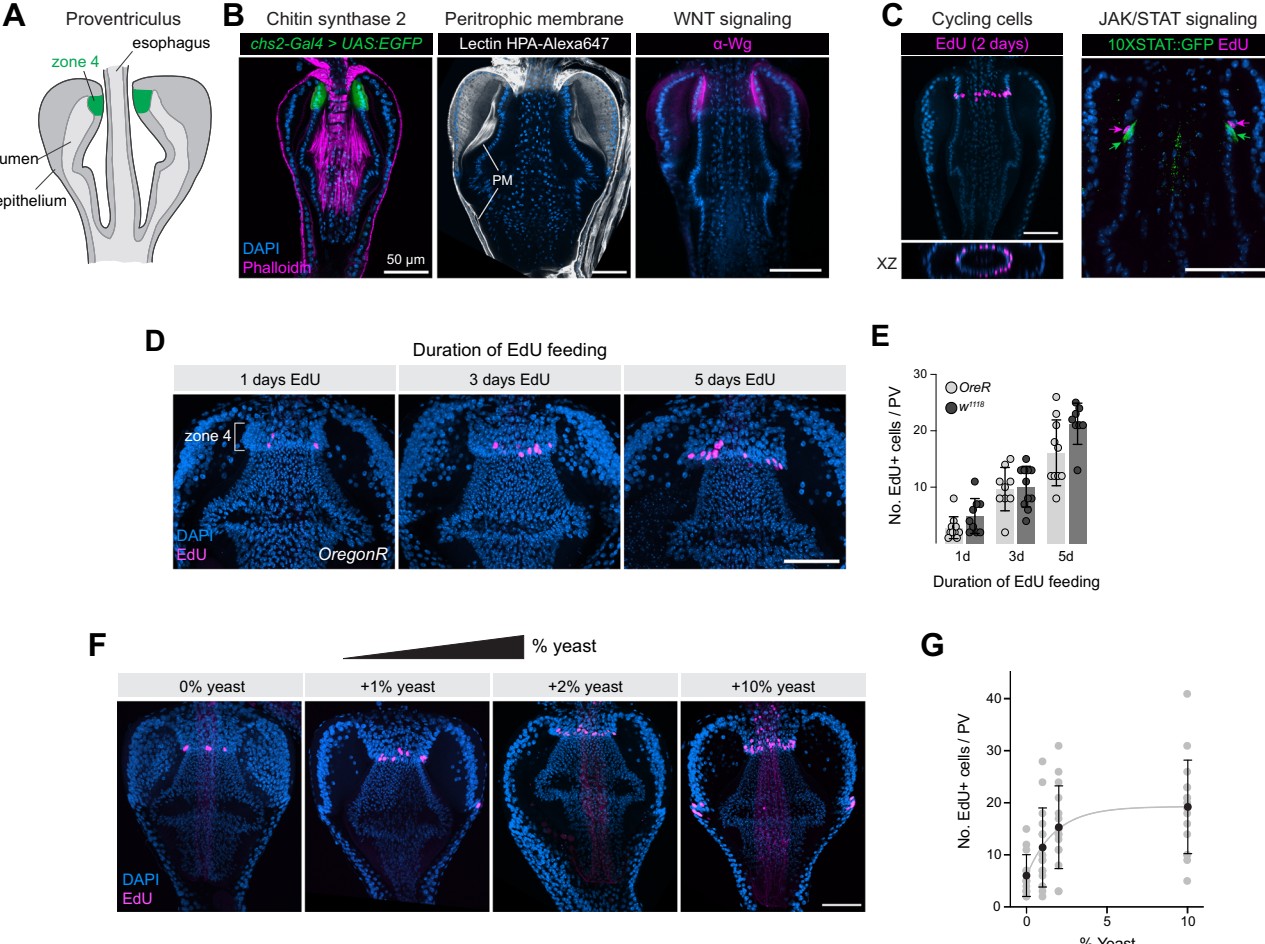

**Fig. 1 | The *Drosophila* proventriculus contains a ring of nutrition-sensitive EdU+ cells located in Zone 4. A** Schematic of the *Drosophila* proventriculus. Zone 4 is indicated in green. **B** Left: *Chs2-Gal4* is expressed in Zone 4 cells. Representative image is shown from N = 2 independent experiments. Middle: HPA lectin stains the peritrophic membrane (PM). Representative images are shown from >10 independent experiments. Right: Wg protein is enriched in Zone 4 cells. Confocal z-slices are shown. Representative images are shown from >5 independent experiments. **C** Left: EdU labeling is enriched in a ring of cells at the posterior of Zone 4. A maximum intensity projection of EdU staining is overlaid on a single z-slice of DAPI staining. Representative image is shown from >20 independent stainings. Right: EdU+ cells are immediately anterior to cells expressing *10XSTAT:GFP* reporter. Single z-slice is shown. Representative image of a single experiment is shown. **D** The number of EdU+ cells increases with duration of EdU

feeding. Maximum intensity projections are shown. **E** Quantification of EdU+ cell numbers at 1, 3, and 5 days of feeding, for two different fly strains, *OregonR* and *w[1118]*. N = number of independent proventriculi [1, 3, 5 days]: *OreR:* 11, 9, 10; *w[1118]:* 9, 11, 8. Differences between genotypes at each time point are not significantly different from each other, tested using the unpaired two-sided t-test with Welch correction. Data are presented as mean ± SD. **F** The number of EdU+ cells increases as more yeast is added to the food source. Maximum intensity projections are shown. **G** Quantification of the experiment shown in (**F**). N = number of independent proventriculi [0, 1, 2, 10%]: 14, 15, 15, 16. Black dots represent mean values, error bars indicate SD. **F** The data were fitted using a one-phase association model, yielding a rate constant K = 0.5689 (95% CI = 0.1639–1.463). Anterior is up, scale bar is 50 μm in all images. Source data are provided as a Source Data file.

the peritrophic membrane, a chitin- and mucin-rich structure that lines the midgut[16–18]. Studies in other Dipterans have shown that the proventriculus is capable of producing peritrophic membrane at a rate exceeding 6 mm/h at 25 °C, representing an enormous capacity for synthesis and secretion[22,23]. In addition to peritrophic membrane synthesis, the proventriculus plays important roles in the immune system[24,25], and as a site for regulation of the microbiota[26].

Morphological studies have identified six distinct "zones" arranged linearly along the proventriculus epithelium, based upon cellular morphology and on their proposed role in the synthesis of the peritrophic membrane[16]. These anatomical zones have been shown via single cell RNAseq (scRNAseq) to correspond to transcriptionally distinct cell types[25].

Zone 4 is a ring-shaped structure that encircles the esophagus at the anterior of the proventriculus (Fig. 1A). A previous study identified a ring of cells located at the posterior limit of Zone 4 which consistently incorporate BrdU, indicating that these cycles progress through S phase in adult flies[15]. This study proposed that these cycling cells are multipotent stem cells that proliferate and replace dying cells elsewhere in the proventriculus[15]. An alternative hypothesis, not explored by Singh et al.[15], is that these cells are endocycling and thus undergoing repeated S phases without mitosis.

Functionally, Zone 4 cells are highly secretory and play an important role in the synthesis of chitin and other peritrophic membrane components[16,25,27]. scRNAseq studies indicate it is enriched for genes involved in chitin synthesis and chitin modifications, including *chitin synthase 2 (Chs2)*[25] which can be visualized with *Chs2-Gal4 > UAS:2XEGFP* (Fig. 1B, left). Studies in larval flies confirm that mutations in *Chs2* lead to defects in peritrophic membrane synthesis[27], and visualizing the peritrophic membrane with fluorescently labeled *Helix pomatia* (HPA) lectin[28,29] confirms previous observations that peritrophic membrane components is highly enriched in the lumen directly abutting Zone 4 cells (Fig. 1B, middle.) Zone 4 cells also express high levels of the WNT pathway ligand Wingless (Fig. 1C, right), as previously reported[15].

To re-investigate the function of these cycling cells, we fed flies with EdU for 2 days and confirmed the presence of a ring of EdU+ cells at the posterior edge of Zone 4 (Fig. 1C, left). EdU+ cells are occasionally observed elsewhere in the proventriculus (see e.g., Figs. 1F and 2A), but here we focus exclusively on the EdU+ population in Zone 4. Previous studies suggested that, in addition to Wg expression, JAK-STAT signaling is also enriched in this area[15], and when we co-labeled proventriculi for EdU and *10XSTAT:GFP*, a JAK-STAT pathway reporter, we found that the EdU+ population is directly anterior to STAT+ cells, suggesting that EdU+ cells are enriched at the posterior edge of Zone 4, with *10XSTAT:GFP* expression representing the anterior limit of Zone 3 (Fig. 1C, right).

To characterize the dynamics of EdU incorporation over time, we fed flies with EdU for 1, 3, or 5 days. We performed this experiment with two standard lab genotypes, *w1118* and *OreR*, and observed that the number of labeled cells increased linearly with time (Fig. 1D, E; simple linear regression, *p < 0.0001* that slope is non-zero for both genotypes) at a rate of between 3.3 (*w1118*) and 4.1 (*OreR*) EdU+ cells per day, suggesting that cells in this region are continually entering S phase at a relatively constant rate. The number of EdU+ cells detected for these two genotypes were not significantly from one another at each time point (Fig. 1E).

To test whether individual cells cycle continuously over time or whether they cease cycling after a discrete number of S phases, we performed a pulse-chase experiment by feeding EdU for 3 days, followed by a 20 day chase in the absence of EdU (Supplementary Fig. 1). After 20 days, we observed significantly fewer in EdU+ cells in Zone 4, suggesting that the EdU signal was diluted by continued cycling (Supplementary Fig. 1). This dilution over time is similar to what is observed for the continually cycling ISCs, in which EdU labeling is diluted through mitosis, and in enterocytes (ECs), which endocycle and thereby dilute the proportion of Edu-labeled DNA in each nucleus over time (Supplementary Fig. 1). In contrast, these results differ from what is observed in the hindgut, where BrdU signal is retained for 20 days due to a lack of continual cycling under homeostasis[11].

It has previously been shown that systemic nutritional signals tightly correlate with endocycle progression in a variety of cellular contexts in the *Drosophila* larva, whereas mitotic activity is not clearly linked to nutritional status[30,31]. We tested whether the EdU+ cells in Zone 4 are responsive to nutrition by varying the amount of dry active yeast, a primary source of protein and other macronutrients for *Drosophila*, added to a standard minimal lab food. We found that the number of EdU+ cells steadily increased in response to additional yeast added to the fly's food (Fig. 1F, G; one-phase association model, rate constant $K = 0.5689$ [95% CI = 0.1639–1.463]), suggesting that the cycling of these cells is influenced by systemic nutritional cues.

We further tested this sensitivity to systemic nutrition using a genetic tumor model in the midgut (*esg-Gal4, tubGal80ts > UAS:yki3SA*; note that *esg-Gal4* is not expressed in the proventriculus), which promotes systemic muscle wasting and cachexia via the upregulation of the systemic insulin antagonist *Impl2*, the cytokine *upd3*, and other cytokines[32–34]. In cachectic flies, we observed a significant decrease in the number of EdU+ cells in the proventriuclus (Supplementary Fig. 2A, B), suggesting that cycling in these cells responds to systemic insulin signaling. We then tested whether over-expressing the insulin antagonists *Impl2*, *upd3*, or both together, would lead to a similar reduction in EdU+ cells in the proventriculus. Indeed, when we expressed these factors from ISCs using *esg-Gal4*, we observed a significant decrease in EdU+ cells of the proventriculus (Supplementary Fig. 2C). Previous studies have shown that, in addition to ISCs, *esg-Gal4* is also expressed in the corpus allatum, located near the proventriculus. We thus used an independent Gal4, *dMef2-Gal4*, to over-express secreted factors from the muscles, and observed a significant decrease in EdU+ cells in the proventriculus (Supplementary Fig. 2D).

Altogether, these observations confirm that the adult proventriculus contains a ring of cells at the posterior of Zone 4 which consistently cycle through S phase, and that they do so in response to systemic nutritional signals.

## Single nucleus atlas and Gal4 toolkit for the proventriculus

In order to functionally characterize these EdU+ cells in vivo, we generated a toolkit of Gal4 lines to label and manipulate various cell types of the proventriculus, building on a recent single cell atlas generated for this tissue[25]. First, we screened the K-Gut database[35] for Gal4 lines that specifically label Zone 4. We identified four lines that labeled Zone 4, inclusive of EdU+ cells, (*GMR41F06-GAL4, GMR26D04-GAL4, GMR56D02-GAL4, and GMR71F06-Gal4*) and selected a particularly specific and consistent line, *71F06-Gal4*, for further study (Fig. 2A).

Next, to identify molecular markers for other proventriculus cell types and to serve as a foundation for further molecular study, we created a single cell transcriptional atlas of the proventriculus using single nucleus RNAseq (snRNAseq). We sequenced a total of 22,332 nuclei from approximately 600 dissected proventriculus tissues, which ultimately resolved into 14 transcriptionally-defined clusters (Fig. 2B–D). This atlas is available via an online data portal (https://www.flyrnai.org/scRNA/proventriculus/).

We mapped the identity of these snRNAseq clusters to in vivo anatomy using cluster-specific marker genes. One cluster was enriched for molecular markers of muscles (e.g., *Mhc, up*) and two clusters were enriched for pan-neuronal markers (e.g., *nSyb, elav, brp*), one of which was enriched for *Akh*- and *Impl2*-expressing cells and therefore likely represents the corpora cardiaca[36–38]. To map the remaining 11 clusters to the in vivo proventriculus, we examined the expression of top marker genes for each cluster using either Gal4 lines or split-intein Gal4 lines crossed to a ubiquitous reporter (Fig. 2C–E)[39].

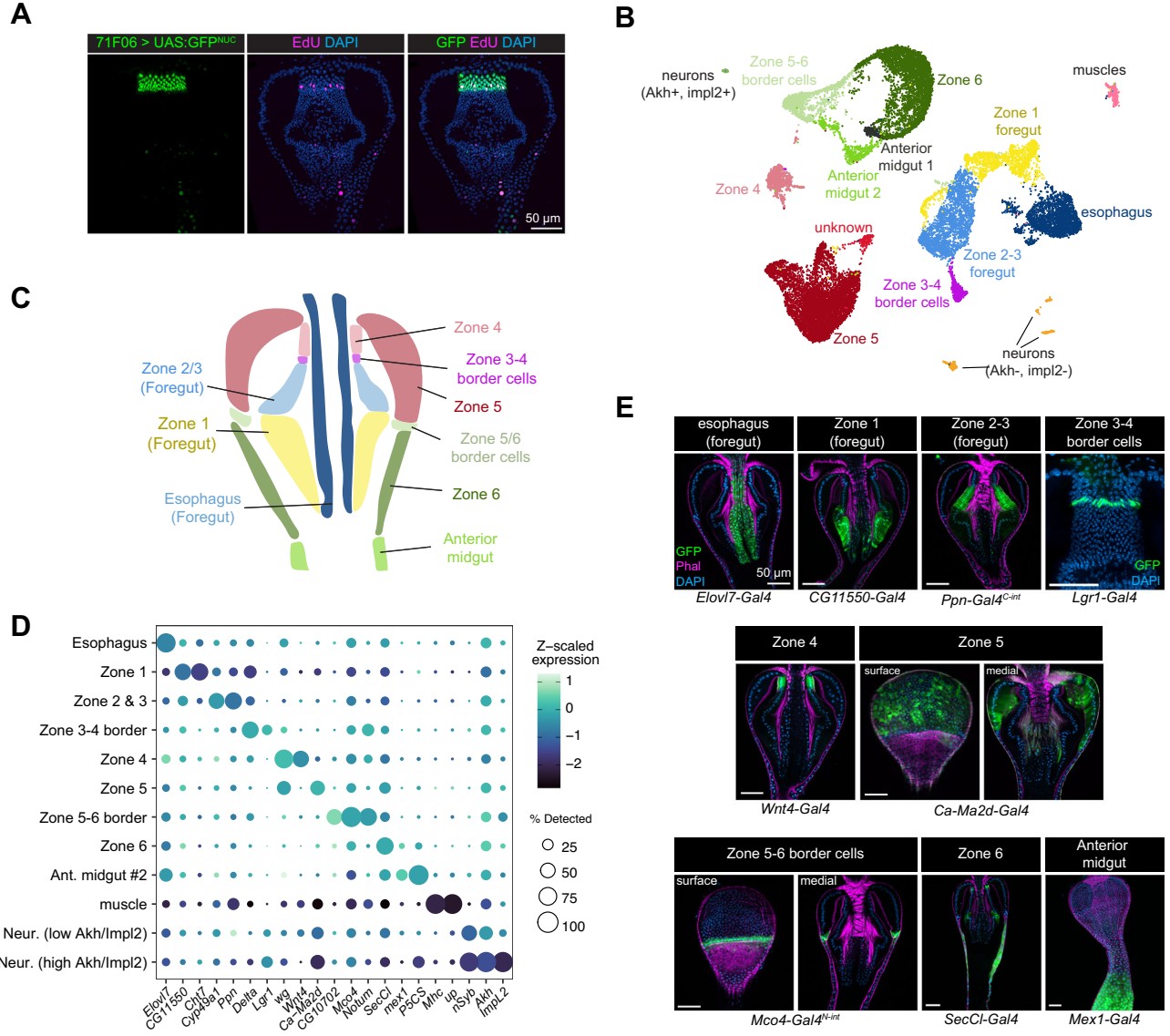

**Fig. 2 | Molecular markers and a single-nuclei atlas for the proventriculus.**
**A** *71F06-Gal4* drives specific expression in Zone 4 cells, including EdU+ cells. Representative image is shown from > 10 independent stainings. **B** UMAP projection of a single snRNAseq dataset created from 22,332 nuclei, containing 14 clusters representing the major epithelial, muscle, and neuronal cells found in the proventriculus. **C** Schematic of the major epithelial cell types in the proventriculus,

corresponding to the clusters identified in (**B**). **D** Dot plot of selected marker genes for the 12 major clusters identified via snRNAseq; note that two minor clusters ("anterior midgut 1" and "unknown") are excluded from this figure as they could not be clearly linked to in vivo cell types. **E** Expression patterns of marker genes for each of the 9 major epithelial cell types in the proventriculus. Anterior is up. Confocal z-slices are shown.

We were able to identify the remaining clusters as anatomically distinct epithelial cell types. One cluster corresponded to the esophagus (marked by *Elovl7-Gal4*), another to Zone 1 (marked by *CG11550-Gal4* and *Cht7-Gal4*). At the clustering resolution we selected, Zone 2 and 3 cells were both found in a single cluster (enriched for *Ppn-Gal4* and *Cyp49a1-Gal4*). We observed a distinct cluster enriched for *Lgr1* and *Delta* that corresponds to a population of cells specifically at the anterior border of Zone 3, immediately posterior to Zone 4. We recovered distinct clusters corresponding to Zone 4 (*Wnt4-Gal4*) and Zone 5 (*Ca-Ma2d-Gal4*), as well as a cluster that corresponds to a discrete band of cells at the widest diameter of the proventriculus, at the border of Zone 5 and 6 (marked by *CG10702-Gal4*, *Notum-Gal4*, *Mco4-Gal4^{N-int}*). A cluster enriched for *SecCl-Gal4* corresponded to Zone 6, and a cluster enriched *mex1-Gal4* and *P5CS-Gal4* corresponded to the anterior midgut. Two additional minor clusters (labeled "unknown" and "anterior midgut 1" in Fig. 2B) were enriched for markers

corresponding to multiple cell types and were excluded from further analysis.

We queried our atlas for markers of stem cells. In a previous scRNAseq atlas of the *Drosophila* midgut[40], midgut ISC clusters were enriched for the stem cell marker *esg*, as well as cell-cycle related genes *PCNA* and *stg*. We did not observe notable expression of any of these genes, nor any cluster-specific enrichment for any *Cyclin* genes, in our proventriculus atlas, whether in our Zone 4 cluster or elsewhere.

Our snRNAseq atlas is largely concordant with a recently published scRNAseq atlas of the proventriculus[25], with minor deviations in the proportions of cells present in each cluster (Supplementary Fig. 3). This high degree of concordance suggests that the cell types we identify are robust to experimental approach, as these two atlases were created using entirely different single cell protols; Zhu (2024) sequenced single cells from freshly dissected and chemically dissociated suspensions, while we sequenced isolated nuclei from flash

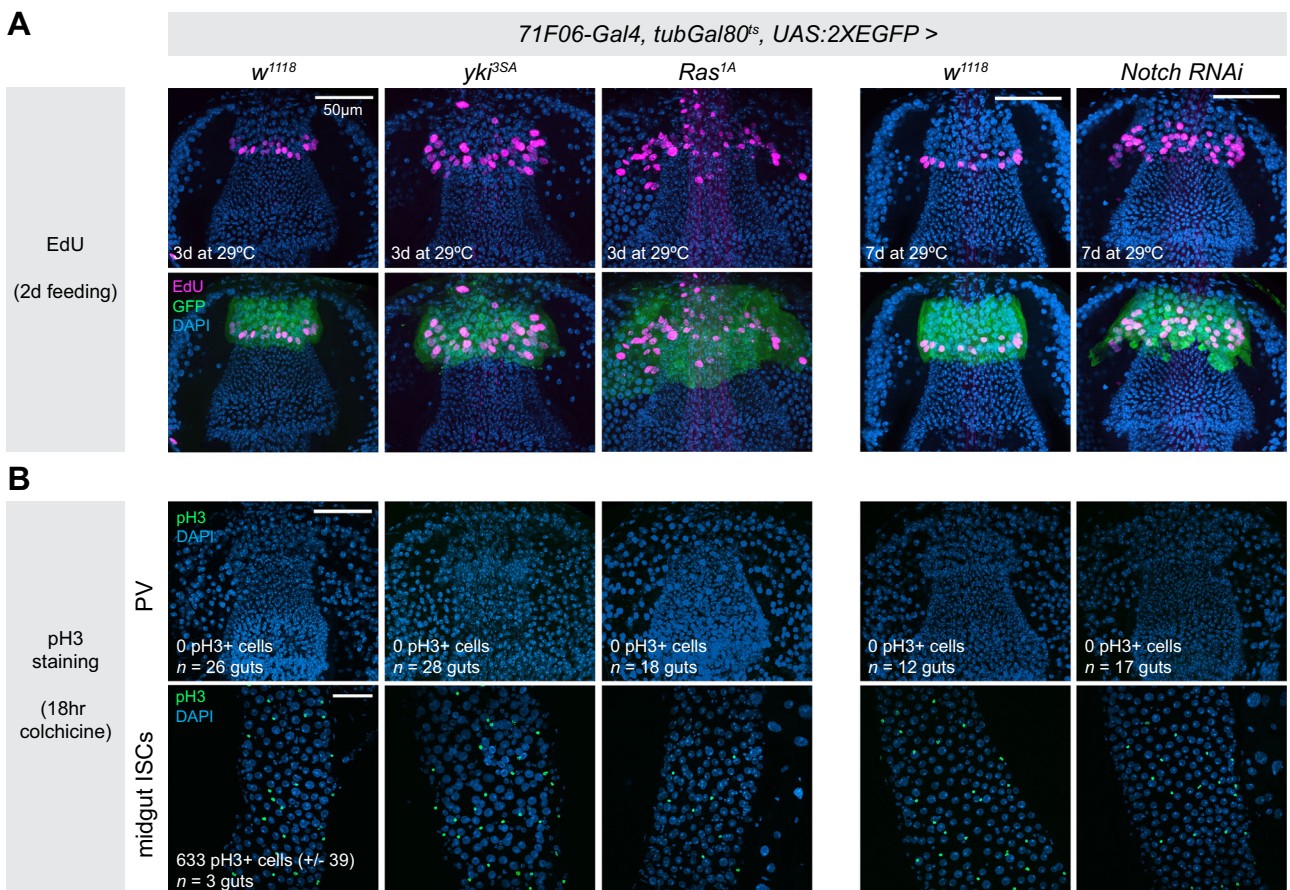

**Fig. 3 | Mitosis is undetectable in Zone 4 cells. A** Over-expression of *yki^{3SA}*, *Ras^{1A}*, or *Notch-RNAi* in Zone 4 cells increases the number of EdU+ cells and enlarges the size of Zone 4. These data are quantified in Supplementary Fig. 4. **B** Zero pH3+ cells were detected in any of the proventriculi examined in any of these genotypes. Flies were fed colchicine for 18 h prior to dissection to enrich for pH3+ cells. Midgut ISCs from these same guts were included as positive controls, which contained many pH3+ staining in all samples. *N* = number of independent guts, as listed in the figure. Maximum intensity projections are shown. Anterior is up, scale bar is 50 μm in all images. Source data are provided as a Source Data file.

frozen tissues. We also note that our atlas was generated from two biological samples: proventriculi from control flies and from cachectic flies, and thus includes cells across a range of nutritional status (see "Methods"). This snRNAseq atlas thus serves as a resource for studies of the proventriculus, and allows us to identify cell type-specific Gal4 driver lines for functional experiments.

**No evidence for mitosis in the proventriculus**

Stem cells are defined as multipotent precursor cells that undergo repeated asymmetric mitosis, giving rise to one daughter cell that retains a stem identity and another daughter cell capable of differentiation. Thus, mitosis is a critical feature of any stem cell system. A previous study of the proventriculus suggested that rare mitoses do occur in this tissue, albeit at a rate that is seemingly out of proportion with the steady numbers of cells passing through S phase[15].

To re-examine whether these cells undergo mitosis, we first stained guts from two different control genotypes (20 *OreR* wildtype guts, 36 *71F06-Gal4 > UAS:2XEGFP* guts, total *n* = 56 guts) for phosphorylated Histone 3 (pH3+), a marker of mitosis. We fed these flies colchicine overnight prior to staining, which enriches for pH3+ cells by blocking exit from mitosis[11]. We observed zero pH3+ cells in the epithelia of Zone 4 in these flies, or anywhere else in the proventriculus.

The cell cycle can be experimentally driven forward using a variety of genetic manipulations, and in cells that are capable of mitosis these manipulations will dramatically increase M phase. For example, in ISCs of the posterior midgut, over-expression of *UAS-yki^{3SA}* [41–43] or *UAS-Ras^{1A}* [44–46], or *Notch-RNAi* [4,5] all lead to dramatic increases in mitosis, leading to massive tumor growth. We hypothesized that, if Zone 4 cells are competent to undergo mitosis, overexpression of such cell cycle-promoting factors should lead to an increase in mitotic cells.

To test whether these factors can drive mitotic proliferation in the proventriculus, we used *71F06-Gal4^{ts}* to drive *UAS-yki^{3SA}*, *UAS-Ras1^{A}*, or *Notch-RNAi* in Zone 4 cells, and then stained for pH3 as a marker of mitosis following enrichment via colchicine (here and below, *71F06-Gal4^{TS}* refers to flies containing both the *71F06-Gal4* driver and *tubGal80^{ts}*, to restrict Gal4 function to adult flies).

All of these genetic manipulations led to significant increases in the number of EdU+ cells and the size of Zone 4 measured as the cross-sectional area at the tissue's widest point (Figs. 3, S4A–D and S4F–H). Under these staining conditions, we could detect over 600 mitotic ISCs in the midgut that stained positively for pH3 (Fig. 3B). In contrast, we observed zero pH3+ cells in the proventriculus under any of these conditions (Fig. 3B). Interestingly, however, we did observe a significant increase in the ploidy of Zone 4 nuclei following over-expression of both *UAS-yki^{3SA}* and *UAS-Ras1^{A}*, indicating that these manipulations promote endocycling (Supplementary Fig. 4E and see below.) In contrast, while *Notch-RNAi* did increase the number of EdU+ cells, it did not increase ploidy, suggesting *Notch* signaling in the proventriculus may function to restrict inappropriate entry into S phase and/or endocycle progression, but not completion of the endocycle.

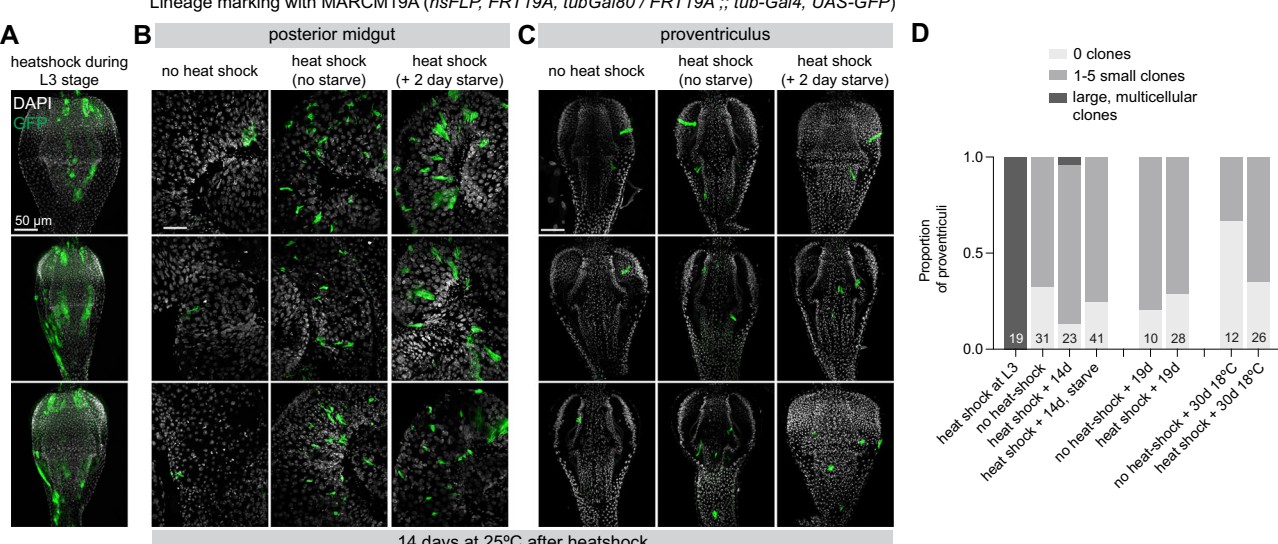

Fig. 4 | **Clonal analysis with MARCM does not label large clones in Zone 4.**
**A** Positive control for clone induction. Clones were induced during the L3 larval stage, leading to large GFP+ clones throughout the proventriculus, including in Zone 4. **B** In the posterior midgut, clone induction labels ISCs and their descendants, generating GFP+ clones throughout the midgut. These clones are generally larger when flies endure a 2-day starvation followed by re-feeding. **C** In the proventriculus, small scattered GFP+ cells are detected throughout the tissue, but no discernable enrichment in Zone 4, where EdU+ cells are located. **D** Quantification of several MARCM experiments, with dissection 14 days after clone induction (at 25 °C), 19 days after clone induction (at 25 °C), or 30 days after clone induction (at 18 °C). Maximum intensity projections are shown. *N* = number of independent guts, as shown in the figure. Data represent multiple examples from single MARCM experiment, with sample sizes given in (**D**). Scale bar is 50 μm. Anterior is up. Source data are provided as a Source Data file.

These results demonstrate that EdU+ cells located in Zone 4 of the proventriculus do not undergo mitosis either under homeostatic conditions or when experimentally driven through cell cycle progression via ectopic oncogene expression. Furthermore, we note that despite morphological deformities caused by over-expressing activated Yki or Ras1, we never saw GFP+ cells located outside of Zone 4, which provides further evidence that cycling cells in Zone 4 do not migrate to replace damaged or dead cells in other regions of the tissue as suggested by Singh et al. [15].

## No evidence for clonal growth of Zone 4 cells

Despite the absence of pH3 staining, it is formally possible that rare mitoses occur. Another hallmark of adult stem cells is the ability to generate multipotent clones in the adult organism. For example, in the adult posterior midgut, *MARCM*-based lineage tracing of ISCs leads to the presence of GFP-labeled clones throughout the midgut that contain all of the epithelial cell types that can be generated by ISC differentiation[4,5]. We therefore used *MARCM*-based lineage tracing to test whether there is clonal growth in the adult proventriculus.

As a positive control for our labeling protocol, we heat-shocked *MARCM* larvae during the L3 larval stage, when this tissue is still undergoing developmental proliferation. We observed large GFP+ clones throughout the proventriculus, both in Zone 4 and elsewhere, in every sample examined, demonstrating that *MARCM* labeling functions as expected in this tissue (Fig. 4A). As a negative control, we examined GFP+ patterns in both midguts and proventriculi from non heat-shocked flies, and observed small numbers of scattered GFP+ cells throughout the proventriculus, as well as in the midgut. We scored proventriculi as having either no GFP+ clones, between 1–5 small groups of GFP+ cells (typically 1–2 cells each), or having large GFP+ clones that include >10 cells (Fig. 4D).

To test for clonal growth during the adult stage, we heat-shocked *MARCM* flies several days after eclosion, then reared them for varying lengths of time before scoring for GFP+ clones. We challenged an additional cohort of flies with a 2-day starvation followed by re-feeding, which increases ISC proliferation in the midgut[47,48].

As expected, in the posterior midgut we observed GFP+ clones in the midguts of heat-shocked flies after 14 days, and we also found that these clones tended to be larger in flies that experienced a 2 day starvation (Fig. 4B). In contrast, in the proventriculus of these flies, we observed scattered singlet and doublet GFP+ cells throughout the tissue at roughly similar rates between negative controls and heat-shocked flies (Fig. 4C, D). This was true at 14 days after heat-shock, 19 days after heat-shock, and 30 days at 18 °C after heat-shock (Fig. 4C, D). We did observe a single example of a large clone in one cohort of heat-shocked animals (Fig. 4D), which was likely a rare example of a *MARCM* clone being formed via leaky FLP expression during earlier development.

In the midgut, ISCs divisions increase dramatically upon tissue damage[6,7]. To test whether there may be damage-responsive proliferation in the proventriculus, we repeated these *MARCM* experiments in flies subjected to a variety of gut damage conditions: bleomycin treatment, infection with pathogenic *ECC15* bacteria, and 2000 RADs of X-rays[6,7]. We also performed *MARCM* experiments with 10% yeast supplementation because we had observed that EdU incorporation rates increased with yeast supplementation. We did not observe discernable GFP+ clones emanating from Zone 4 or any other area of the proventriculus in any of these conditions (Supplementary Fig. 5). We note the caveat that, unlike enterocytes of the midgut, Zone 4 cells do not directly contact the contents gut lumen and therefore likely do not experience the same degree of damage that ECs experience in contact with bleomycin or *ECC15*. Altogether, we find no evidence for clonal growth of the EdU+ cells in Zone 4 of the proventriculus.

## No proliferation in response to distant tissue damage

Singh et al. proposed that the EdU+ cells in the proventriculus are proliferating for the purpose of migrating to replace dying cells elsewhere in the tissue[15]. This hypothesis implies that cells originating in Zone 4 proliferate in response to cell death in nearby tissues, and then migrate to such nearby tissues as the esophagus, anterior midgut, and crop.

To directly test how the EdU+ cells of Zone 4 respond to cell death in nearby tissues, we used the Gal4-UAS system to induce cell death in various regions of the proventriculus by overexpressing the proapoptotic factor *UAS-rpr*. We induced cell death in three different cell types of the proventriculus: in Zone 5 (*Ca-Ma2d-Gal4*), in Zone 1 (*CG11550-Gal4*), and in the esophagus (*Evolv7-Gal4*) (Fig. 4). In these experiments, we silenced *UAS-rpr* throughout development using *tubGal80^{ts}* and raised flies at 18 °C until several days post-eclosion. We activated tissue damage by shifting adult flies to 29 °C for 48 h, and placed flies on EdU-containing food for the duration of the tissue damage, and throughout a recovery phase of 2 days or 5 days at 18 °C following damage. If these EdU+ cells in Zone 4 proliferate to compensate for dying cells, we would expect to see an increase in EdU+ cells in response to damage in neighboring cells, and likely EdU+ cells migrating towards the area of damage.

In all three experiments, we observed pyknotic nuclei, indicative of apoptosis, in the intended cell type in *UAS-rpr* flies, and not in control flies (Fig. 5A–C), indicating that this system induces apoptosis as expected. However, in all three experiments we observed not an increase in EdU+ cells, but rather a significant decrease (Fig. 5.) These results are inconsistent with the hypothesis that these cells proliferate in response to tissue damage elsewhere in the proventriculus. The decrease in EdU+ cells supports our previous observation that these cells cycle in response to cues indicative of organismal health.

Interestingly, when we drove apoptosis in the esophagus with *Elovl7-Gal4*, while we observed a decrease in EdU+ cells in Zone 4, we did observe indications of massive endoreplication in the esophagus itself: unusually large, EdU+ nuclei (Fig. 5C, asterisks). This is similar to what has been reported in the hindgut[11]. In contrast, tissue damage to Zone 1 or Zone 5 did not cause an obvious increase in endoreplication in the damaged tissue (Fig. 5A, B).

## Genetic lineage tracing of EdU+ cells in Zone 4

As a final test for lineage tracing independent of MARCM, we wished to genetically label the descendants of Zone 4 EdU+ cells using a Gal4 line specific to these cycling cells. However, we did not observe any discernible transcriptional signatures of the Zone 4 EdU+ cells, and were initially unable to identify a Gal4 driver to specifically label these cells within Zone 4.

We thus used the split-intein Gal4 system to generate a genetic driver that specifically labels the EdU+ population of cells in Zone 4[39]. We generated a split-intein Gal4^{C-int} line using the 71F06 enhancer fragment (Supplementary Fig. 6A; referred to here as *71F06^{Gal4-C-int}*), and then used our snRNAseq atlas to identify *Delta* as a candidate gene which is primarily enriched in Zones 1–3 and in Zone 3–4 border cells, with some expression as well in Zone 4 (Supplementary Fig. 6A). When we examined the intersectional labeling of *71F06^{Gal4-C-int}* ∩ *Delta^{Gal4-N-int}* > *UAS:2xEGFP*, we observed specific expression in a band of cells at the posterior edge of Zone 4 (Supplementary Fig. 6A). We confirmed that these cells contain the large majority of EdU+ cells (Supplementary Fig. 6A).

To lineage trace these cells, we used the G-TRACE system combined with *tubGal80^{ts}*, which labels current expression in RFP and lineage expression in GFP[49]. After development at 18 °C, we placed *71F06^{Gal4-C-int}* ∩ *Delta^{Gal4-N-int}, tubGal80^{ts}* > G-TRACE at 29 °C to initiate lineage tracing and examined the proventriculus after 1 day, 6 days, and 10 days (Fig. 6C, D). If these cells are proliferating and generating clones, we would expect to see progressively more GFP+/ RFP− cells, as descendants of these cells would grow into the rest of the tissue. However, while we did observe small numbers of GFP+/RFP− cells immediately adjacent to the GFP+/RFP+ domain, the number of these cells did not increase or move over time, but were constant over a 10 day period (Fig. 6C, D). These results are consistent with leaky and/ or minor dynamic changes in the expression pattern of our split-intein Gal4 line, rather than with proliferation of cells in this domain.

Taken together, we find no evidence that the EdU+ cells located in Zone 4 are stem cells. We do not observe mitosis, nor the generation of MARCM clones, nor a proliferative response to nearby tissue damage, nor any evidence of genetic cell lineages growing over time. We thus propose that these cells must be endocycling rather than undergoing mitosis.

## Cells in Zone 4 increase in ploidy as flies age

The consistency with which EdU is incorporated into cells in Zone 4 suggests that these cells are continually endocycling throughout the fly's life, which should lead to increased ploidy as flies age. To test this directly, we performed a time-course experiment using confocal microscopy to quantify the ploidy of Zone 4 nuclei over the course of fly aging. In this experiment, *71F06-Gal4 > UAS:GFP^{stinger}* flies (hereafter, *UAS:GFP^{NUC}*), which express nuclear-localized GFP in Zone 4, were split into vials either containing dried active yeast powder *ad libitum* or standard food lacking additional yeast powder, and were examined at six timepoints between 3 days and 35 days after eclosion. At each timepoint, we collected 5-6 confocal z-stacks and quantified the ploidy and the volume of these nuclei (Supplementary Fig. 7).

We observed a consistent increase in ploidy and nuclear volume over the course of the fly's life, supporting the hypothesis that these cells are continually endocycling and increasing in ploidy (Figs. 6A–C and S7A, B; relationship between age and ploidy was significantly different than zero; simple linear regression; $n = 960/1064$ cells[no yeast/yeast], $p < 0.0001$ for both no yeast and yeast).

Consistent with the observation that supplemental yeast increases EdU incorporation, we found that the 7 and 14 day timepoints, ploidy was significantly higher in flies which received dried active yeast, but that by 21 days the average ploidy was not significantly different between these treatments (Fig. 6A–C). Over this time-course, the mean ploidy of Zone 4 cells increased from ~4 C at 3 days (2.1–2.2 × diploid, with or without yeast) to ~10 C at 35 days (4.8–4.9 × diploid), while the maximum ploidy amongst Zone 4 cells increased from ~9 C (4.3–4.6 × diploid) to ~22–26 C (11.1–13.0 × diploid) between these time points. We found no evidence that Zone 4 cells become multinucleate, based on co-staining these samples with the f-actin marker, even at the latest time point (35 d) when the tissue begins to lose a clearly monolayered structure (Supplementary Fig. 6C).

## Zone 4 cells are progressively lost into the gut lumen

Surprisingly, we observed that while the volume of Zone 4 nuclei progressively increased, the total number of GFP+ nuclei steadily decreased (Fig. 6D). Over the first 14 days, cell numbers decreased at equivalent rates for flies fed with or without yeast, but beginning at 21 days the number fell faster for flies without added yeast (Fig. 6D). Remarkably, we could observe "snap shots" of this cell loss in fixed samples, visible as GFP+ nuclei being shed into the gut lumen at the time of dissection (Figs. 6E and S7). We quantified the number of GFP+ nuclei present in the lumen outside the Zone 4 epithelium in these fixed tissues and found that the number of GFP+ cells outside of Zone 4 first started to increase at 14 days after eclosion, and subsequently increased until reaching a maximum at 28 days (Fig. 6F). We did not observe any obvious morphological differences in the cells being lost, and the mechanism by which these cells exit the epithelium is not known, but we observed holes forming in the F-actin-rich barrier separating Zone 4 cells from the gut lumen, with GFP+ nuclei passing through these holes into the lumen (Supplementary Fig. 8). We note that the patterns of cell loss neatly with our total cell counts, with the greatest decreases in cell numbers corresponding to those time-points when more nuclei were being lost.

Together, these data suggest that Zone 4 nuclei are endocycling and thus increasing in ploidy and nuclear volume while cells from this region are being lost into the lumen. A simple explanation for these observations is that ploidy is increasing to maintain

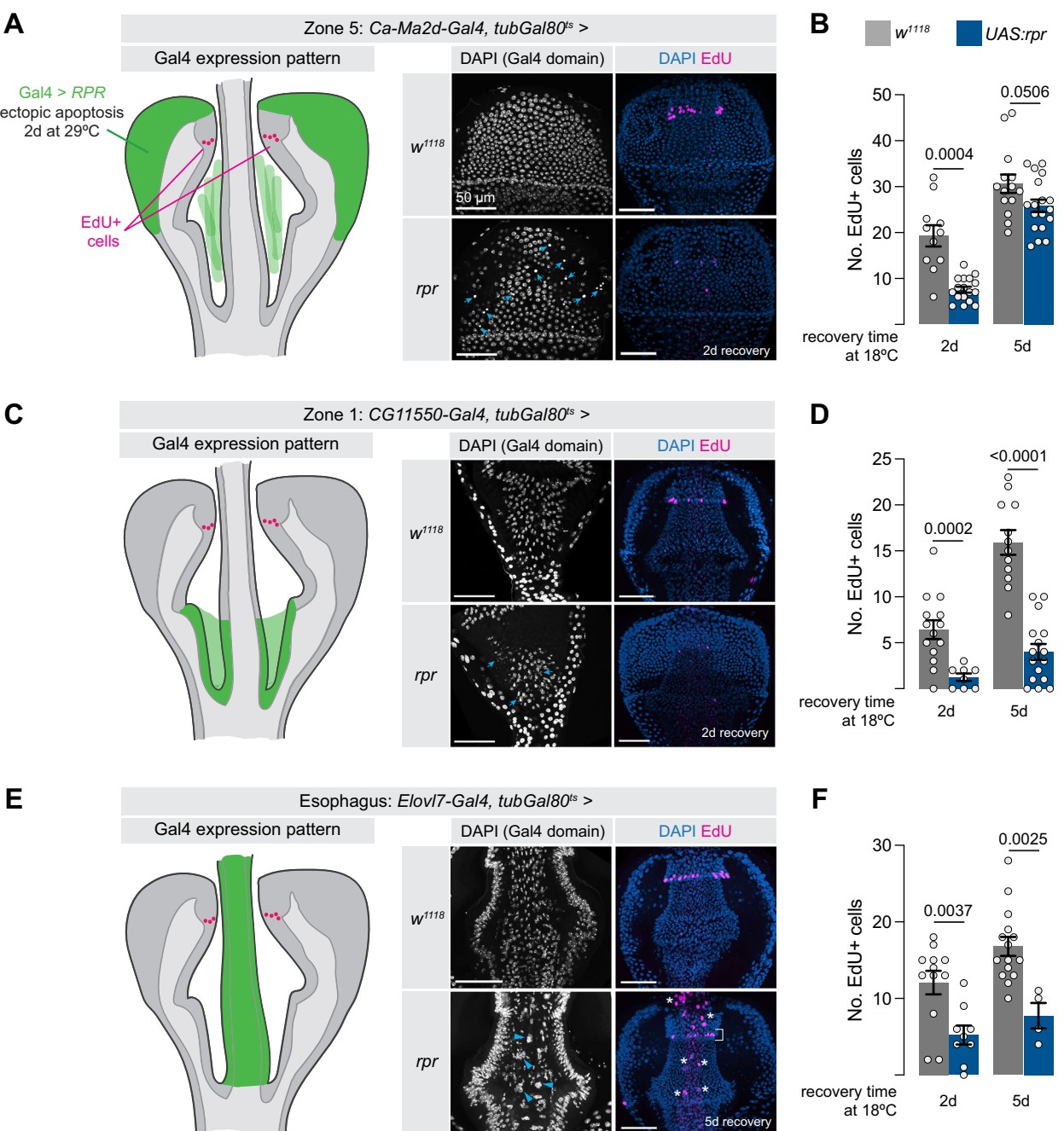

**Fig. 5 | Zone 4 cells do not proliferate to replace cells lost elsewhere in the proventriculus.** Tissue damage was induced in three distinct cell types of the proventriculus by over-expressing the pro-apoptotic gene *rpr* for 2 days, and EdU+ cells were visualized after two and 5 days of recovery. **A** *rpr* expression in Zone 5, using *Ca-Ma2d-Gal4*, causes a decrease in EdU+ cells after both 2 and 5 days. Blue arrows indicate pyknotic nuclei indicative of cell death. **B** Quantification of experiment shown in (**A**) *N* = number of independent proventriculi [2, 5 days]: *w*[1118]: 11, 14; *rpr*: 17,18. **C** *rpr* expression in Zone 1, using *CG11550-Gal4*, causes a decrease in EdU+ cells after both 2 and 5 days. **D** Quantification of experiment shown in (**C**).

*N* = number of independent proventriculi [2, 5 days]: *w*[1118]: 14, 12; *rpr*: 8,16. **E** *rpr* expression in the esophagus, using *ELOVL7-Gal4*, leads to decreased EdU+ cells in Zone 4, and endoreplication in esophageal cells. White bracket indicates EdU+ signal in Zone 4, blue arrowheads and white asterisks indicate large, EdU+ nuclei in the esophagus itself. **F** Quantification of experiment shown in (**E**). *N* = number of independent proventriculi [2, 5 days]: *w*[1118]: 12, 15; *rpr*: 9,4. Statistical tests in (**B**, **D**, **F**) are two-sided unpaired t-tests, with Welch's correction for unequal variances for the 2 day tests in (**B**) and (**D**). Data are presented as mean ± SEM. Anterior is up. Scale bar is 50 μm in all images. Source data are provided as a Source Data file.

metabolic output and tissue integrity as cells are progressively lost during aging.

### Tissue damage within Zone 4 leads to endocycling

Given our observation that Zone 4 cells endocycle at the same that cells are being lost from the tissue, a simple hypothesis is that endocycling is triggered by the loss of cells from Zone 4, as a compensatory measure. Alternatively, the opposite could be true: as endocycling increases tissue size, this increased tissue size drives the loss of cells to maintain tissue size.

To distinguish between these possibilities, we returned to the genetic cell ablation strategy described above, this time focusing on Zone 4 itself. We raised *71F06-Gal4*[ts] > *rpr* and control flies (*71F06-Gal4*[ts] > +; Fig. 7A) at 18 °C until 2–3 days post-eclosion, then shifted

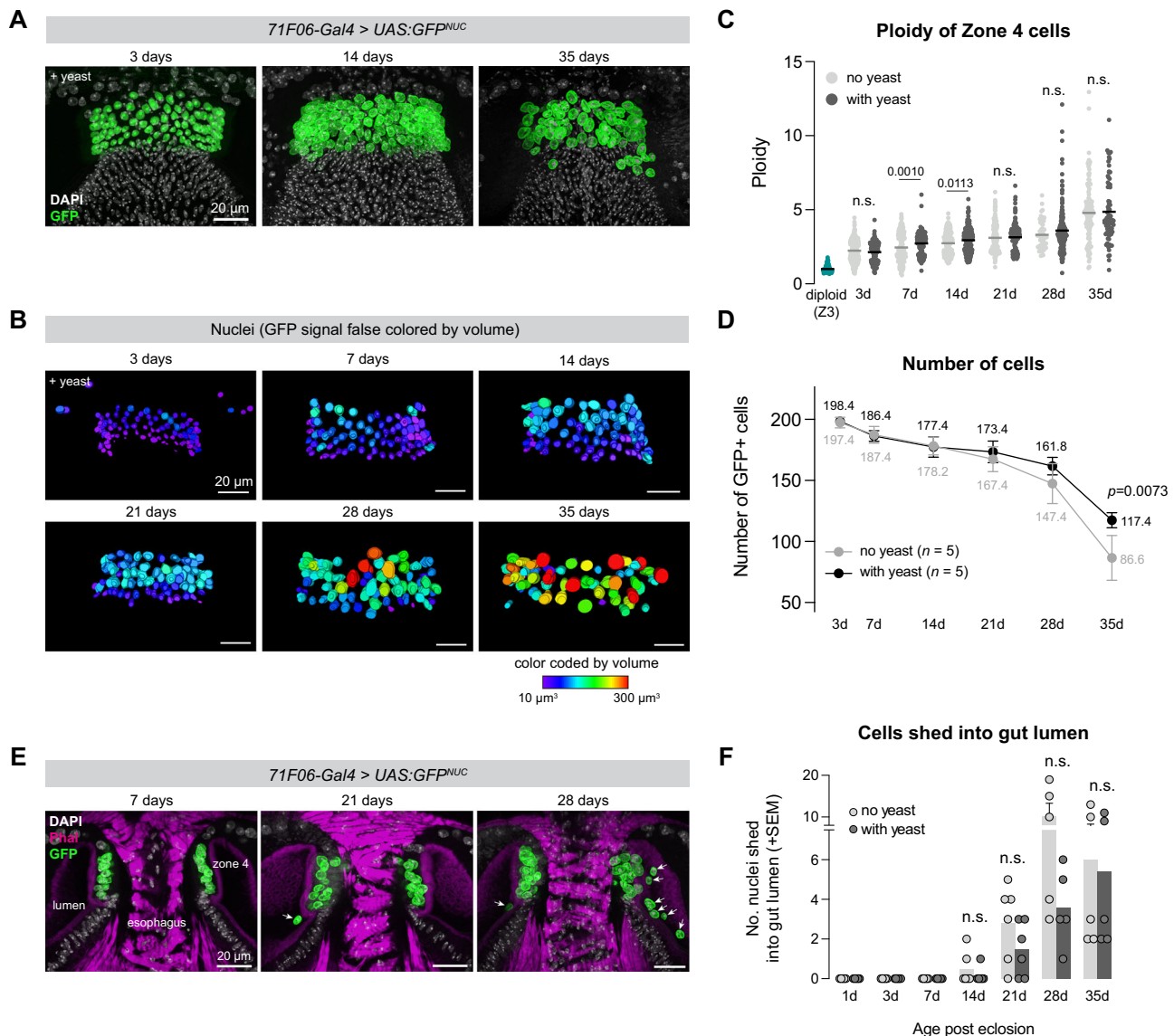

**Fig. 6 | Zone 4 ploidy increases over the fly lifespan, while cells are continually lost into the gut lumen during aging. A** Time-course of Zone 4 ploidy over the course of the lifespan of the fly. Nuclear-localized GFP was expressed in Zone 4 using *71F06-Gal4*, and flies were provided with or without supplemental yeast powder. **B** Nuclei false-colored by volume. **C** Quantification of ploidy of Zone 4 nuclei, normalized to diploid cells nuclei from Zone 3 cells in the same images. Each dot represents a single nuclei, and lines indicate mean. Ploidy increases throughout the lifespan of a fly, and is significantly higher in yeast-fed flies at the 7 and 14 day time points. Statistical tests are two-sided unpaired *t*-test, with Welch's correction for unequal variance for the 7 and 28 day timepoints; "n.s." = not significantly different. *N* = number of nuclei measured (no yeast, with yeast): diploid: 102, 3 d: 244,

217, 7 d: 206, 198, 14 d: 174, 228; 21 d: 176, 175; 28 d: 52, 168; 35 d: 108, 78. Repeated measurements were made on nuclei from N = 5 independent proventriculi, except for *N* = 6 for 14 day yeast sample and *N* = 2 for 28D no yeast sample. **D** The number of Zone 4 cells steadily decreases throughout between 3 days and 35 days post-eclosion, and decreases faster in the absence of yeast powder. Data are shown as mean ±± SEM. *N* = 5 independent proventriculi per datapoint. Statistical tests were two-sided unpaired *t*-tests. **E** GFP+ nuclei are consistently observed shed into the lumen adjacent to Zone 4 (**F**) Quantification of GFP+ cells observed in the lumen at each time point. Data are shown as mean ± SEM. *N* = 5 independent proventriculi per datapoint except *N* = 6 for 14 and 21 day samples, and for 3 days+ yeast sample. Statistical tests were unpaired *t*-tests. Source data are provided as a Source Data file.

them to 29 °C for either 2 or 5 days to induce cell death in Zone 4. During this damage period, flies were fed with EdU-containing food to monitor cell cycling in response to tissue damage. We observed small, pycnotic nuclei throughout Zone 4, indicative of cells in Zone 4 (Fig. 7B, arrowheads).

After 2 days of *rpr* expression in Zone 4 the number of EdU+ cells decreased significantly compared to control flies, but after 5 days this number had rebounded to control levels (Fig. 7C). Notably, the ploidy of these EdU+ cells following 5 days of tissue damage was significantly increased (Fig. 7D). Thus, when cell loss is ectopically induced in Zone 4 via tissue injury, the remaining cells in the tissue endocycle to

increase their ploidy. These results are consistent with the hypothesis that Zone 4 cells detect the loss of nearby cells and increase their ploidy via the endocycle in response.

These results indicate that Zone 4 cells can endocycle in response to exogenous damage, yet our previous observations suggest that cells in this tissue also endocycle continuously throughout the fly's life even in the absence of exogenous damage (Figs. 1 and 7). Specifically, we observe continuous endocycling during the first two weeks of the fly's life when there is only a modest amount of cell loss (Fig. 7D). To reconcile these observations, we hypothesized that there are two phases of endocycling in this tissue: first, in young flies, endocycling

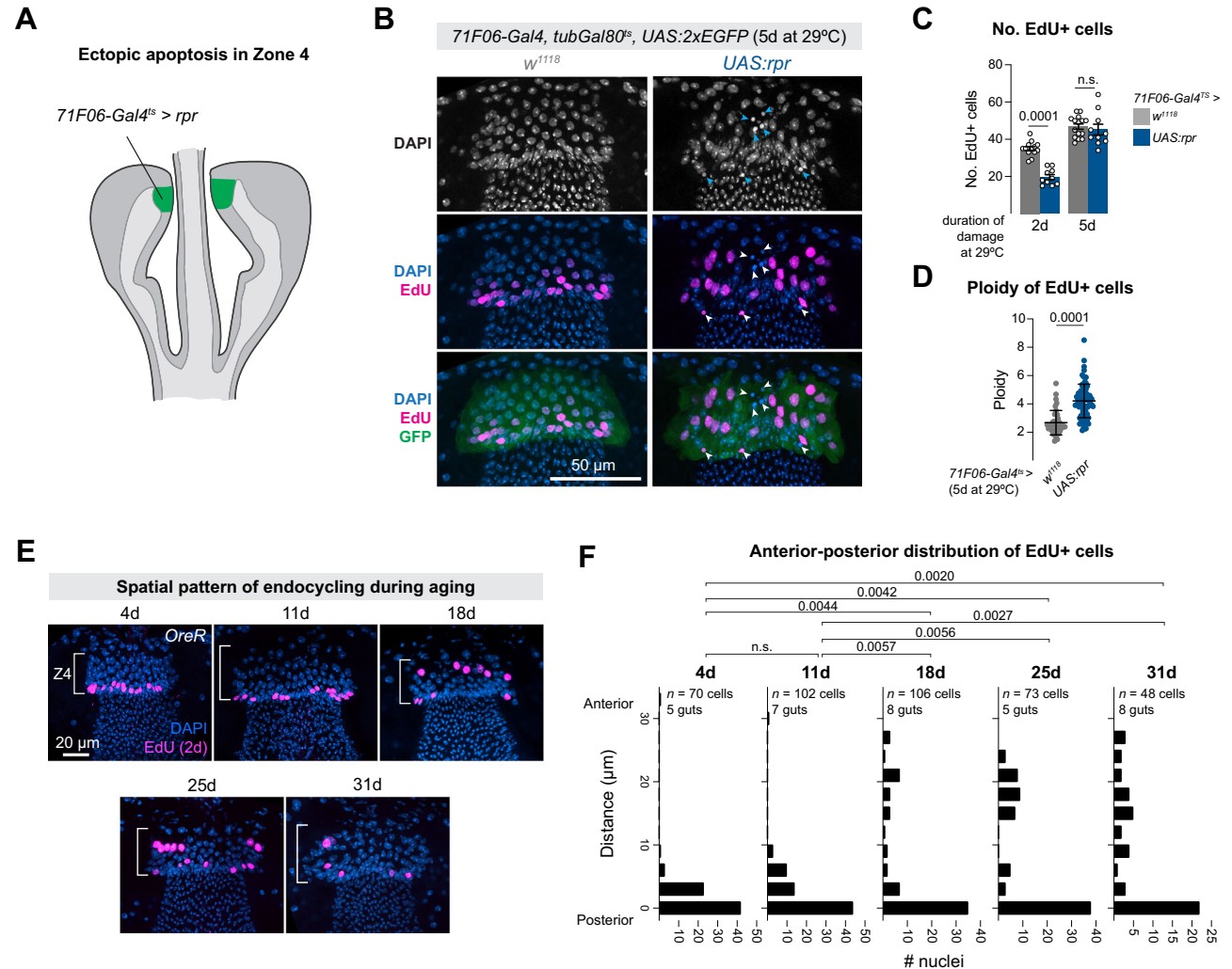

**Fig. 7 | Loss of cells from Zone 4 induces nearby endocycling. A** *71F06-Gal4^ts > rpr*
drives apoptosis in Zone 4, for either 2 or 5 days. **B** Top row: pycnotic nuclei in Zone
4 indicate cell death (arrowheads). Middle and bottom rows: following tissue
damage, EdU+ cells in Zone 4 appear larger are located throughout the tissue rather
than enriched at the posterior of Zone 4. Images show tissues after 5 days of *rpr*
expression. **C** The number of EdU+ cells decreases after 2 days or *UAS:rpr* expres-
sion, but recovers to wildtype levels after 5 days of damage. *N* = number of inde-
pendent proventriculi (*w^{1118}*, *rpr*): 2 day: 13, 10; 5 day: 15, 10. Mean and SEM are
shown. Statistical tests are two-way unpaired t-tests. **D** The ploidy of EdU+ cells is
increased following tissue damage in Zone 4. *N* = number of EdU+ nuclei scored:

*w^{1118}*: 39; *UAS:rpr*: 79, repeated measurements from 3 independent *w^{1118}* proven-
triculi and 6 *UAS:rpr* proventriculi. Mean ± SD are shown. Statistical test is two-way
unpaired t-test with Welch correction. **E** Spatial pattern of EdU+ cells in Zone 4 in
aging flies. In younger flies (4 and 11 day old), EdU+ cells are restricted to the
posterior edge of Zone 4, whereas in older flies (18, 25, 31 days), EdU+ cells are
found throughout more anterior portions of Zone 4. **F** Histograms of the distance
of EdU+ nuclei from the posterior edge of the tissue. *N* = number of EdU+ nuclei and
independent proventriculi, as shown in the figure. Statistical tests are
Krustal–Wallis tests. Source data are provided as a Source Data file.

occurs in the absence of cell loss, and is spatially restricted to the
posterior edge of the tissue. Subsequently, when naturally occurring
cell loss from Zone 4 begins to accelerate in ~2–3 week old flies, a
second phase of endocycling begins in response to cell loss, and is
spatially distributed throughout Zone 4, in areas nearby the sites of
cell loss.

A prediction of this hypothesis is that the spatial distribution of
EdU+ cells should be markedly different in the proventriculi of young
vs old flies: restricted to the posterior in young flies, and distributed
throughout the tissue in older flies. To test this, we collected *OreR* flies
at five timepoints between 4 and 31 days old, and measured the spatial
distribution of EdU+ cells relative to the posterior edge of Zone 4.
Remarkably, in younger flies (4 and 11 days), EdU+ cells were almost
entirely restricted to 1–2 cell diameters from posterior of the tissue
(Fig. 7E, F). However, in all three of the older time-points (18, 25,
31 days), in addition to those EdU+ cells located at the posterior of the
tissue, there were also many EdU+ cells scattered throughout Zone 4,

and these distributions were significantly different from the younger
time-points (Fig. 7E, F). These results suggest that, in wildtype flies
under unperturbed conditions, there are two distinct phases of
endocycling in the proventriculus: a steady, continuous endocycling in
younger flies at the posterior of the tissue, and subsequently
throughout the tissue in response to increased cell loss as the fly ages.

**Excess endocycling increases peritrophic membrane synthesis**
Increasing genomic copy number is a widespread mechanism to
enhance synthesis of proteins and other biological components in
non-proliferative tissues[31,50,51]. Given the role that Zone 4 cells play in
the synthesis and secretion of chitin and other peritrophic membrane
components[25,27], we hypothesized that endocycling in this tissue is
linked to peritrophic membrane synthesis. We therefore used genetic
manipulations to both increase and decrease endocycling and
then visualized the peritrophic membrane and tested for functional
defects.

To increase endocycling, we overexpressed the cell cycle regulators *UAS:CycE* or *UAS:Myc* in Zone 4 cells using *71F06-Gal4ts*. CycE is a key regulator of both endocycling and the mitotic cell cycle, and drives entry into S phase[31,52], and Myc is a transcription factor whose overexpression is sufficient to drive endocycling and increased nuclear volume in multiple contexts[31,51,53,54]. In Zone 4 cells, both manipulations led to significantly enlarged nuclei after 7–9 days at 29 °C (Fig. 8A, B), as well as significantly increased ploidy (Fig. 8F), indicative of increased endocycling. As predicted, these manipulations also led to significant increase in the tissue size of Zone 4 (Fig. 8G).

We tested whether increased ploidy led to altered peritrophic membrane synthesis by staining with a panel of molecular markers. Confocal projections of HPA lectin-stained tissues revealed large areas of glycosylated material emanating from Zone 4, forming thickened cords that were not observed in controls (Fig. 8B). To quantify this phenotype, we measured the cross-sectional area of HPA lectin-stained material adjacent to Zone 4 and found a significant increase in *UAS:CycE* and *UAS:Myc* flies (Fig. 8C, H). We then stained these tissues for calcofluor, which labels chitin, a major component of the layer of peritrophic membrane synthesized by Zone 4. We observed a significant increase in calcofluor-positive signal adjacent to Zone 4 in *UAS:CycE* and *UAS:Myc* flies, indicative of increased chitin synthesis (Fig. 8D, I). Lastly, we stained these tissues with wheat germ agglutinin (WGA), an additional marker for glycosylated targets, and which has been used as a peritrophic membrane marker[55], and observed a significant increase in WGA staining adjacent to Zone 4 as well (Fig. 8E, J). Together, these results suggest that increasing endocycling in Zone 4 leads to increased synthesis of peritrophic membrane in the proventriculus.

These experiments with *UAS-CycE* and *UAS-Myc* also allowed us to perform an independent test of whether endocycling was functionally causal for the loss of cells from Zone 4 during aging. If cell loss was driven by the process of tissue growth via endocycling, then experimentally increasing endocycling via *UAS-Myc* or *UAS-CycE* should lead to enhanced cell loss. We counted the number of GFP+ cells in these genotypes 1 day after shifting to 29 °C, and after 12 days (Supplementary Fig. 9). The number of cells in each genotype decreased after 12 days by between 21 and 31%, but the percent decrease between genotypes were not significantly different (Supplementary Fig. 9), despite the fact that ploidy, nuclear size, and tissue size is significantly increased in these genotypes compared to controls (Fig. 8). This suggests that the loss of Zone 4 cells during aging is not driven by endocycling itself, and that a distinct mechanism must govern the loss of cells.

**Blocking endocycling reduces peritrophic membrane synthesis**
To inhibit endocycling, we used RNAi to knock down several candidate genes implicated in the regulation of the endocycle in *Drosophila*: *fizzy-related (fzr)*, *E2F1*, *myc*, *CycE*, and the insulin receptor (*InR*)[12,56–64]. Of these candidates, *InR-RNAi* was by far the most potent reagent to block endocycling and reduce ploidy in Zone 4 (Supplementary Fig. 10 and Fig. 9). *fzr-RNAi* did not reduce ploidy (Supplementary Fig. 10A), *E2F1-RNAi* caused a statistically significant yet relatively modest reduction in ploidy (Supplementary Fig. 10B), and *myc-RNAi* led to dramatic disruptions to Zone 4 structure beyond simply blocking endocycling (Supplementary Fig. 10C). While *cycE-RNAi* did reduce ploidy and tissue size (Supplementary Fig. 10D, F, G), this effect was muted compared to *InR-RNAi* and caused large aggregations of peritrophic membrane material that complicated further analysis (Supplementary Fig. 10D, E). Thus, we focused our analysis of inhibited endocycling to *InR-RNAi*, and we note the caveat that Insulin signaling has additional effects of cellular metabolism in addition to simply regulating the cell cycle.

Using two independent RNAi constructs, we observed a near total loss of EdU staining in *71F06-Gal4ts > InR-RNAi* animals (Fig. 9A, B) and a

concordant failure to increase ploidy during aging (Fig. 9C). In control flies (*71F06-Gal4ts > w1118* or *mCherry-RNAi*), Zone 4 nuclei steadily increase in ploidy between 6 days and 21 days at 29 °C, where in *71F06-Gal4ts > InR-RNAi* flies, ploidy did not increase (Fig. 9C). The overall size of Zone 4, visualized with cytoplasmic *UAS:2XEGFP*, was dramatically reduced by *InR-RNAi* (Fig. 9D, H). These experiments also demonstrate that the link between systemic nutrition and endocycling in Zone 4 (e.g., Fig. 1F, G) is at least partially regulated cell autonomously via insulin signaling directly interpreted by these cells.

We visualized the peritrophic membrane in *71F06-Gal4ts > InR-RNAi* flies using the same three molecular markers as above. Consistent with the reduced nuclear volume and tissue size, the quantity of peritrophic membrane emanating from Zone 4 in *InR-RNAi* animals was significantly reduced, as revealed by staining for HPA lectin, calcofluor, and WGA (Fig. 9E–G, I–K). These alterations in peritrophic membrane morphology are consistent with the hypothesis that increased genomic copies allow for increased synthesis of peritrophic membrane components.

The peritrophic membrane is continuously synthesized in the proventriculus, then migrates posteriorly to line the midgut, where it serves as a protective barrier for the absorptive surface of enterocytes[65,66]. To assess the ultimate functional consequences of reduced endocycling in Zone 4 cells, we performed a number of assays to characterize the structure and function of the peritrophic membrane in the midguts of *71F06-Gal4ts > InR-RNAi* flies. When viewed using transmission electron microscopy (TEM), the peritrophic membrane can be seen to contain four distinct layers[16]. Zone 4 is thought to synthesize the contents the second layer, L2, which is chitin-rich and appears as a thick, electron-dense layer abutting the lumen[16] (Supplementary Fig. 11A). We analyzed the peritrophic membrane of *InR-RNAi* flies using TEM (14 days at 29 °C), and did not observe any gross morphological defects or alterations to the four-layered structure (Supplementary Fig. 11A), indicating that L2-containing peritrophic membrane is still produced in flies with reduced endocycling in Zone 4.

We then performed two independent assays for intestinal barrier integrity on *71F06-Gal4ts > InR-RNAi* flies. First, we subjected *71F06-Gal4ts > InR-RNAi* flies to the "smurf assay", in which flies are fed a blue dye which leaks from the gut into the rest of the body in flies with a compromised gut barrier[67,68]. In flies which had been aged for 24 days at 29 °C, we did not observe any gut leakage either in controls or *71F06-Gal4ts > InR-RNAi* flies (Supplementary Fig. 11B), indicating that these guts retained a baseline of functional integrity. Similarly, when we fed flies with fluorescently-labeled beads (diameter 0.056 μm) to test for leakage[55,66], we did not detect a statistically significant increase in the proportion of *71F06-Gal4ts > InR-RNAi* flies with leaky peritrophic membranes compared to controls (Supplementary Fig. 11C). These two assays suggest that flies with inhibited endocycling in the proventriculus are still capable of producing sufficient peritrophic membrane to function under standard laboratory conditions.

The peritrophic membrane is a critical barrier to pathogens[66,69,70], so we hypothesized that reduced endocycling in Zone 4 cells may ultimately lead to functional defects in pathogen resistance. Under standard conditions at 29 °C, the lifespan *71F06-Gal4 > InR-RNAi* flies was indistinguishable from control flies (Fig. 9L), consistent with the above observations that peritrophic membrane is not grossly malformed or dysfunctional. However, when these flies were challenged with the pathogenic bacteria *Pseudomonas aeruginosa* (PA14), *InR-RNAi* flies were significantly more susceptible to infection than controls (Fig. 9M), indicating that the peritrophic membrane has compromised function. These results suggest that while continual endocycling of Zone 4 cells is not strictly required for peritrophic membrane synthesis, it instead provides robustness in the face of environmental challenges such as pathogenic bacteria.

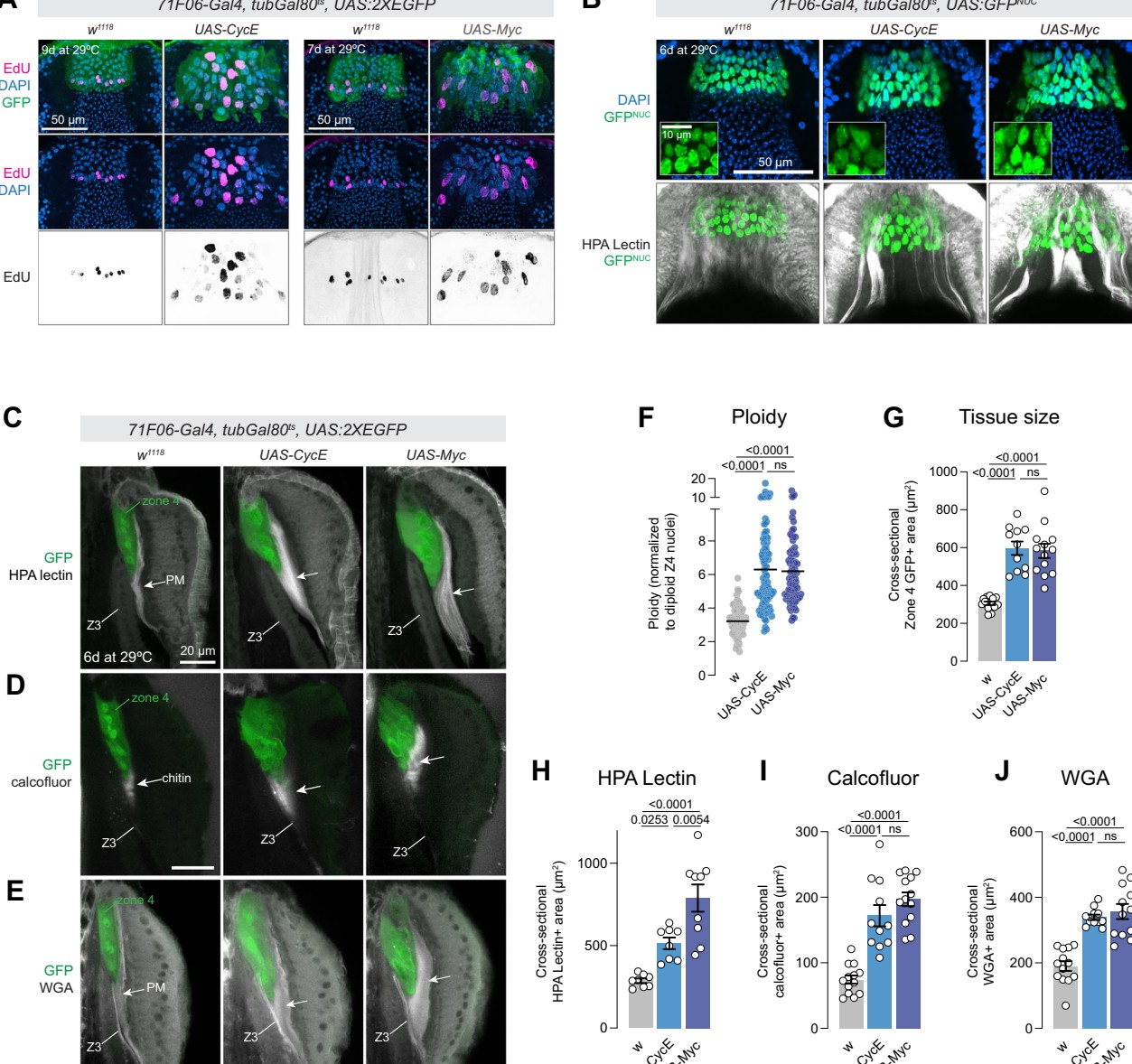

**Fig. 8 | Increased endocycling in Zone 4 cells leads to increased peritrophic membrane synthesis in the proventriculus. A** Over-expression of *CycE* (left) or *Myc* (right) in Zone 4 increases endocycling and nuclear size. Representative images from a single experiment are shown. **B** *71F06-Gal4ts > CycE* or *Myc* increases nuclear size (top) and altered peritrophic membrane synthesis. Insets show increased magnification. **C–E** *71F06-Gal4ts CycE* or *Myc* leads to increased tissue size, and (**C**) increased HPA lectin staining; (**D**) increased calcofluor staining, and (**E**) increased amount of WGA staining. Images are maximum intensity projections of 3-4 z-slices. Z3 = zone 3, which includes a chitinous border that is distinct from peritrophic membrane. **F** Quantification of ploidy for flies shown in (**B**). Each dot represents a nucleus, bar represents the mean. *N* = 85 *w1118* nuclei from 4 proventriculi, 102 *CycE* nuclei from 5 proventriculi, and 88 *Myc* nuclei from 4 proventriculi. Statistical test is a one-way ANOVA with Tukey's multiple comparison test. **G** Quantification of tissue

size. In (**G–J**), each dot represents the average of two measurements per proventriculus. *N* = number of independent proventriculi: *w1118* = 13, *CycE* = 11, *Myc* = 13. Statistical test is a one-way ANOVA with Tukey's multiple comparison test. **H** Quantification of HPA lectin staining. Number of independent proventriculi is 8 *w1118*, 8 *CycE*, and 9 *Myc*. Statistical test is a one-way ANOVA with Tukey's multiple comparison test. **I** Quantification of the area of calcofluor-positive material. Number of independent proventriculi is 13 *w1118*, 11 *CycE*, and 13 *Myc*. Statistical test is a one-way ANOVA with Tukey's multiple comparison test. **J** Quantification of the area of WGA-positive material. Number of independent proventriculi is 13 *w1118*, 11 *CycE*, and 12 *Myc*. Statistical test is a one-way ANOVA with Tukey's multiple comparison test. Data in **G–J** are shown as mean ± SEM. Source data are provided as a Source Data file.

Altogether, these experiments demonstrate that endocycling leads to increased ploidy, tissue size, and synthetic capacity in Zone 4, ultimately increases the production of peritrophic membrane components. We propose that endocycling is a compensatory mechanism to ensure that continued production of sufficient quantities of peritrophic membrane components as cells are lost to aging or tissue damage.

## Discussion

We find no evidence of stem cells in the *Drosophila* proventriculus, in contrast to a previous report[15]. Our experiments demonstrate that the EdU+ cells in Zone 4 do not undergo mitosis, do not produce clonal lineages, do not cycle in response to nearby tissue damage, and instead increase in ploidy throughout adulthood. These results suggest that the primary lines of evidence in support of the stem cell hypothesis

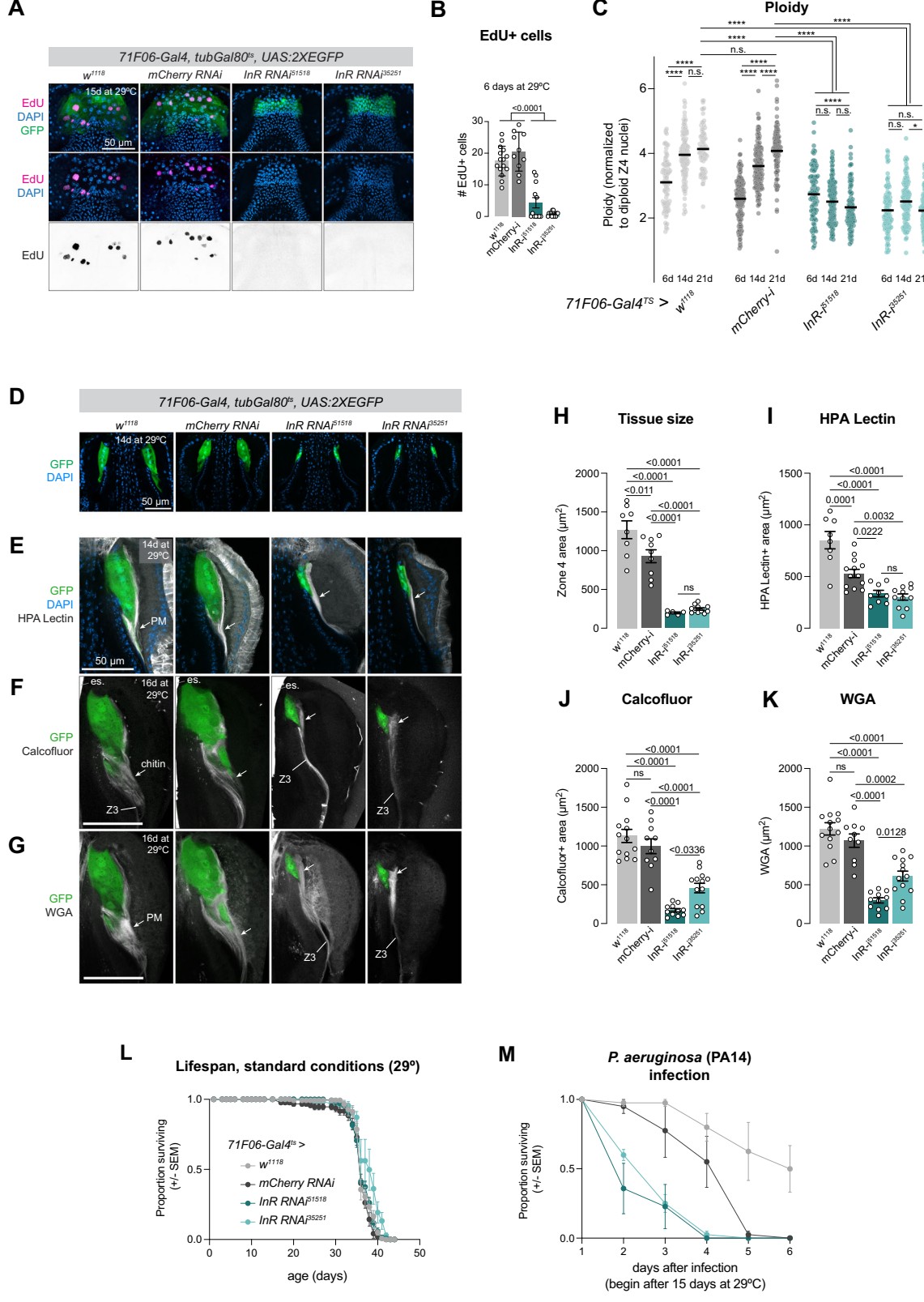

were most likely the result of spontaneous, background MARCM activity as well as leaky expression of the Gal4 line used for lineage tracing[15].

We find that the cells in Zone 4 endocycle continuously throughout the fly's life, even in the absence of exogenous damage, and that a subset of these cells are shed into the gut lumen as the fly ages. We also demonstrate that, causally, Zone 4 cells endocycle in

response to cell loss from the tissue, indicating that the cells can detect nearby cell loss. Functionally, we find that blocking endocycling in Zone 4 causes decreased peritrophic membrane synthesis. Our experiments reveal that flies are remarkably robust in the face of this reduction, and do not show obvious ultrastructural defects or gut leakage. However, flies with endocycle-deficient proventriculi are less resistant to potent bacterial pathogen infection. Thus, we propose that

**Fig. 9 | Blocking endocycling in Zone 4 reduces peritrophic membrane synthesis. A** *71F06-Gal4<sup>ts</sup> > InR-RNAi* reduces EdU incorporation and tissue size. **B** Quantifcation of (**A**). Number of independent proventriculi: 15 *w<sup>1118</sup>*, 10 *mCherry-RNAi*, 12 *InR-RNAi<sup>51518</sup>*, 13 *InR-RNAi<sup>35251</sup>*. Statistical test is Ordinary one-way ANOVA. Data are shown as mean ± SD. **C** *InR-RNAi* reduces ploidy. Bars represent mean. *N* = number of nuclei (6, 14, 21 days) = *w<sup>1118</sup>*: 92, 120, 76; *mCherry-RNAi*: 152, 129, 112, 95; *InR-RNAi<sup>51518</sup>*: 95, 135, 98; *InR-RNAi<sup>35251</sup>*: 105, 121, 120. *N* = number of independent guts (6, 14, 21 days): *w<sup>1118</sup>*: 4, 4, 3; *mCherry-RNAi*: 4, 4, 4; *InR-RNAi<sup>51518</sup>*: 3, 4, 4; *InR-RNAi<sup>35251</sup>*: 4, 4, 4. Statistical test: one-way ordinary ANOVA test with Tukey's multiple comparison test. **** *p < 0.0001*; * *p = 0.0134*. **D** *InR-RNAi* reduces tissue size. *InR-RNAi* reduces peritrophic membrane synthesis: (**E**) HPA lectin, (**F**) calcofluor, and (**G**) WGA. Z3 = zone 3. **H** Quantification of tissue size. In (**H–K**), each dot represents the average of two measurements per proventriculus. *N* = number of independent proventriculi = *w<sup>1118</sup>*: 8, *mCherry-RNAi*: 9, *InR-RNAi<sup>51518</sup>*: 5, *InR-RNAi<sup>35251</sup>*: 11. Statistical

test: one-way ANOVA with Tukey's multiple comparison test. Data in (**H–M**) are presented as mean ± SEM. **I** Quantification of (**E**). *N* = number of independent proventriculi = *w<sup>1118</sup>*: 8, *mCherry-RNAi*: 13, *InR-RNAi<sup>51518</sup>*: 9, *InR-RNAi<sup>35251</sup>*: 11. Statistical test: one-way ANOVA with Tukey's multiple comparison test. **J** Quantification of (**F**). *N* = number of independent proventriculi = *w<sup>1118</sup>*: 13, *mCherry-RNAi*: 10, *InR-RNAi<sup>51518</sup>*: 11 *InR-RNAi<sup>35251</sup>*: 13. Statistical test: one-way ANOVA with Tukey's multiple comparison test. **K** Quantification of (**G**). *N* = number of independent proventriculi = *w<sup>1118</sup>*: 13, *mCherry-RNAi*: 10, *InR-RNAi<sup>51518</sup>*: 11, *InR-RNAi<sup>35251</sup>*: 13. Statistical test: one-way ANOVA with Tukey's multiple comparison test. **L** Lifespan is not affected by *InR-RNAi*. *N* = number of flies: *w* = 134; *mCherry-RNAi* = 95; *InR-RNAi<sup>51518</sup>* = 128; *InR-RNAi<sup>35251</sup>* = 58; 15 flies per vial. **M** *InR-RNAi* leads to increased susceptibility to *PA14*. *N* = number of flies: *w* = 40; *mCherry-RNAi* = 40; *InR-RNAi<sup>51518</sup>* = 38; *InR-RNAi<sup>35251</sup>* = 40; 9–10 flies per vial. Data in (**D–K**) represent single experiments; (**L**) was repeated once with similar results. Source data are provided as a Source Data file.

continual endocycling in Zone 4 ensures sufficient levels of peritrophic membrane synthesis throughout the fly's life to address the diversity of environmental challenges that flies may experience in the wild.

Endoreplication and polyploidy are both widespread in *Drosophila* and across animals and plants, both under healthy, homeostatic conditions and in response to cell or tissue damage[31,50,51,71,72]. In the *Drosophila* larva, many tissues cease mitotic proliferation during embryogenesis, and subsequently grow via developmentally programmed endocycling[31,73]. In adult flies, many cell types become polyploid via endoreplication under wildtype conditions, including the nurse cells and follicle cells of the ovary, the enterocytes of the midgut, and rectal papillae cells[71], while other cell types endocycle in response to tissue or cellular damage, including the epidermis in response to wounding[13,71], the hindgut in response to tissue injury[11,12], and the adult brain in response to oxidative and DNA damage[74]. Mammals also display widespread examples of both developmentally programmed endocycling as well as damage-responsive endocycling, including, for example, in the mammalian liver, which displays high levels of endoreplication in response to damage and injury[31,50,51,71,72].

The pattern of endocycling that we observe in the proventriculus appears to differ in certain respects from several other well-characterized examples of endoreplication in *Drosophila* and other tissues. Unlike ectodermal gut regions, including the hindgut[11,12] or the esophagus (Fig. 5E), endoreplication in the proventriculus appears to occur throughout the fly's life, even in the absence of exogenous tissue injury. Whereas many other highly metabolically active cells become highly polyploid via endoreplication, including the nurse cells or the follicle cells of the ovary, in those cell types, endoreplication appears to proceed for a specific number of cycles and then cease[75], rather than continual endoreplication over the fly's lifespan. What we observe in the proventriculus is a continuous increase in ploidy throughout the adult stage, as part of a homeostatic system that accounts for the steady loss of neighboring cells and responds to the nutritional input of the fly. A similar progressive increase in cell ploidy is seen in the adult brain, where several neuronal and glial cell types that are diploid at eclosion become polyploid during aging in response to DNA damage or oxidative stress[74].

Interestingly, we observe consistent endocycling in young flies, including during the first two weeks of the adult stages (Fig. 6A–C), a period in which we detect very modest age-related cell loss (Fig. 6D). This indicates that endocycling is a consistent feature of this tissue throughout the fly's life, and not solely a response to cell loss. A possible explanation for this consistent endocycling of cells at the posterior edge of Zone 4 is that the production of peritrophic membrane components is intrinsically damaging to these cells, given the very high synthetic and secretory demands of this process. In the midgut, the inherent cellular functions of the enterocytes are thought to cause high levels of cellular stress and thereby require consistent replacement via stem cells, even in the absence of acute tissue damage. Given the absence of stem cells in Zone 4, continual endocycling may serve a

similar function in this tissue, to maintain a functional baseline of synthetic and secretory activity as the tissue accrues damage due to the cellular stress of peritrophic membrane synthesis. There are similar examples of polyploidization in response to cell loss during aging or disease, including in the mammalian cornea, where cell loss due to disease can be compensated for by polyploidization of remaining cells[14].

We propose that there are two distinct stages of endocycling in the *Drosophila* proventriculus: first, during the first ~14 d of a fly's life, regular endocycling at the posterior edge of Zone 4 steadily increases the size of the tissue. Then, as the fly ages and cells are increasingly lost from Zone 4, the surviving cells increase endocycling in order to compensate for the loss of these cells. Interestingly, because the ploidy of these cells steadily increases with age, we propose that it may be possible to estimate the age of a fly based on the distribution of ploidies observed in Zone 4.

Altogether, we propose that this system may provide insights into how the endocycle can be regulated over long timescales, as well as into the mechanisms that link age-related cell loss to endoreplication.

## Methods

### Experimental animals

Genotypes and sources for *Drosophila melanogaster* lines used in this study are provided in Supplementary Table 1. Flies were maintained at 18 °C, 25 °C, and 29 °C, as indicated in the text, on standard laboratory cornmeal food, except for EdU treatments, yeast supplementation experiments, or drug treatments, which were maintained on Formula 4–24 Instant Blue Food ("Instant Blue Food", Carolina Biological Supply) as described below. All experiments were performed on female flies. No ethical approval was required for studies on *Drosophila melanogaster*.

Except where noted in the text, all experiments were initiated on flies within 1 week of eclosion, and all flies were age-matched within an experiment. For temperature shift experiments with *tubGal80<sup>ts</sup>*, flies were reared throughout development at 18 °C and were shifted to 29 °C 1–4 days after eclosing.

### EdU feeding and EdU+ cell counting

EdU (Thermo Fisher) was diluted to 0.2 mM in water, then used to hydrate an equal volume of Carolina Instant Blue Food, supplemented with yeast powder as indicated in the text. Adult flies were transferred to EdU-containing food and flipped to fresh EdU-containing food every 2 days for any experiment lasting longer than 2 days. EdU signal was detected using Click-iT™ EdU Cell Proliferation Kit (Thermo Fisher). Except where noted in the text, 5% yeast was used in EdU experiments (100 mg dried yeast per 2 mL of food), and flies were <2 weeks old. EdU+ cells were manually counted on a compound epifluorescent microscope, specifically focusing on those EdU+ cells located in Zone 4. Unless otherwise noted in the text, flies were fed EdU for 2 days prior to dissection.

## Yeast variation experiments

To test the effect of supplemental dietary yeast on EdU incorporation, dried active yeast was added as a percentage by weight of Instant Blue Food.

## Colchicine treatment

To enrich mitotic nuclei, flies were fed with 0.2 mg/mL colchicine (EMD Millipore) for 12 h prior to dissection. Colchicine was diluted in water and used to dilute Instant Blue Food supplemented with 10% yeast.

## Gut damage with X-rays, bleomycin, and *ECC15* infection

For X-ray damage, adult flies were irradiated with 0, 1000, or 2000 RADs in a TORREX 120D X-ray Inspection System (ScanRay Corporation). For gut damage with bleomycin, bleomycin (Sigma) was diluted to 25 ng/μL in water and used to rehydrate 2 mL of Instant Blue Food. Flies were fed bleomycin-containing food for 2 days at 29 °C to increase ingestion, and were flipped to fresh food each day, with water as a control. For *ECC15* treatment, 100 mL of LB medium was inoculated with 20 μL of a *ECC15* glycerol stock and grown at 30 °C for 20–24 h. *ECC15* cultures were then centrifuged for 30 min at 1810 x *g* and resuspended in 1.4 mL of a 5% sucrose. Flies were starved for 2 h at 29 °C, and then flipped to a vial containing only a kimwipe soaked in 700 μL of the *ECC15* solution, where they fed for 24 h at 29 °C to induce gut damage. After gut damage, flies were maintained on standard food at 25 °C for the indicated amount of time.

## Immunohistochemistry and confocal imaging

Tissues were dissected in 1× phospho-buffered saline (PBS), fixed for 25–30 min in 4% paraformaldehyde in PBS, permeabilized in PBS with 0.3% Triton-X (PBST), and stained following standard antibody staining procedures using 5% normal goat serum as a blocking agent. Filamentous actin was stained using Alexa 555-coupled phalloidin (Molecular Probes), peritrophic membranes were stained with Alexa647-coupled HPA lectin (1:100, Thermo Fisher), calcofluor white stain (1:100, Sigma-Aldrich), Alexa647-coupled wheat germ agglutinin (1:400, Fisher Scientific), and nuclei were stained using DAPI. For HPA Lectin, we found that an incubation of at least two nights at 4 °C, and up to five nights, was necessary for reliable staining, whereas calcofluor and WGA were stained for two nights, and that these peritrophic membrane stainings were improved by isolating the anterior midgut below the proventriculus from the rest of the midgut. Stained tissues were mounted in Vectashield (Vector Labs), except for Alexa647-labeled tissues, which were mounted in Prolong Gold (Thermo Fisher Scientific). Rabbit anti-GFP AlexaFluor488 conjugate (Molecular Probes, 1:300) was used to visualize GFP expression, Wg expression was detected using mouse anti-Wg (Developmental Studies Hybridoma Bank 4D4, 1:100) and phospho-Histone 3 was detected using rabbit anti-PH3 (Cell Signaling Technologies 9701S, 1:500). Primary antibodies were detected with Alexa Fluor 488- or 555- coupled secondary antibodies (1:400). Confocal images were captured on a Zeiss LSM 780, LSM 980, or an Olympus IX83 confocal microscope, through the Microscopy Resources of the North Quad (MicRoN) facility at Harvard Medical School.

## Single-nucleus RNAseq (snRNAseq) atlas

To capture proventriculus transcriptional states in both well-fed flies and flies with reduced systemic insulin signaling, single nuclei suspensions were generated from proventriculus samples from both control flies (*esg-Gal4, tubGal80*[ts]*, UAS:GFP/+*) and cachexic flies (*esg-Gal4, tubGal80*[ts]*, UAS:GFP; UAS-yki*[3SA]), after 8 or 9 days at 29 °C. We verified via confocal analysis that *esg-Gal4* is not expressed in the proventriculus itself, which was also confirmed via snRNA sequencing. Proventriculus samples were manually dissected from adult female flies in cold Schneider's medium and immediately flash frozen on dry ice in Eppendorf tubes until a total of approximately 600 samples were collected (~300 of each genotype). Proventriculi were manually severed from the crop and esophagus, and from the midgut just posterior to the proventriculus, anterior to where *esg-Gal4* is detected in midgut ISCs.

Single nuclei were isolated following the protocol described in ref. [76], derived originally from refs. [77,78], and stained with DAPI immediately before FACS sorting, performed at the Immunology Flow Cytometry Core at Harvard Medical School. Sorted nuclei were resuspended in 1× PBS supplemented with 0.5% BSA with RNAse inhibitor (Promega) at a concentration of ~2000 cells/μl.

snRNAseq was performed following the 10X Genomics Chromium protocol (Chromium Next GEM 551 Single Cell 3'_v3.1_Rev_D), and sequenced using a Illumina NovaSeq 6000 at the Harvard Medical School Biopolymers Facility. 10,525 nuclei from *esg*[ts] flies and 11,801 nuclei from esg[ts] > *yki* flies were sequenced (total = 22,332 nuclei) and reads were mapped to BDGP6.32.

Bioinformatic analysis was performed as follows, and the analysis pipeline is available here: https://github.com/MujeebQadiri/SingleNucleiProventriculusAnalysis/tree/main. Raw data were processed with Cell Ranger software (10x Genomics, v7.0.0) to generate a single-cell matrix for each sample. The count matrices were imported into Seurat (v4.3.0) and cells with less than 1300 or greater than 60,000 UMIs were filtered out, and low-abundance genes with less than 10 counts were removed from further analysis. The two biological samples were bioinformatically pooled for PCA and clustering analysis to generate a single atlas including cells from both conditions. After a PCA reduction, the first 40 components were used for UMAP and clustering analysis, at a 0.2 resolution. We removed from further analysis one cluster that appeared to be an artifact, as it contained markers for multiple different unrelated cell-types as well as stress and heat shock proteins. Differentially expressed genes were calculated using the Wilcox Rank Sum Test across each cluster and were then further selected using a Log2FC > X cut-off and adjusted *p* value < 0.05. All downstream analysis was performed in R (v4.3).

To map snRNAseq clusters to proventriculus anatomy, we examined the expression pattern of Gal4 lines for top-ranking marker genes using confocal microscopy.

## MARCM analysis

Flies of the genotype *hsFLP, FRT19A, tubGal80; tub-Gal4, UAS-GFP* were crossed to *FRT19A* flies (BL1709), and offspring were heat-shocked in a 37 °C water bath either during the L3 stage for during the adult stage. Two 1 h heat-shocks were administered, separated by 2–3 h at 25 °C. MARCM crosses were maintained at either 25 °C or 18 °C as indicated in the text.

## Tissue damage protocol using *UAS-rpr*

We induced ectopic apoptosis by crossing Gal4 drivers to flies containing *UAS-rpr; tubGal80*[ts] or, for non-damaged controls, to *w*[1118] flies. These crosses were maintained at 18 °C until several days after eclosion. Adult flies were then transferred to EdU+ food and placed at 29 °C for 48 h to induce apoptosis. For recovery experiments, flies then returned to recover at 18 °C for either 2 days or 5 days, on EdU+ food throughout. For continuous damage experiments, flies were maintained at 29 °C for either 2 or 5 days. Thus, all EdU+ incorporation events that occurred during the damage or during the recovery period could be visualized and quantified.

## Creation of split-intein Gal4 line

To generate *GMR71F06::Gal4-C-int*, the 3547 bp enhancer fragment from *GMR71F06* FlyLight line[79] was amplified using the following primers: caccTGCCCCGAGCAGCAAAAGAAGTCCG and TCTCCGAAA TCTGCGCCAATTCCAG, and cloned into the pENTR-dTOPO vector (Thermo Fisher Scientific). This fragment was cloned into the

pBP-Gal4[C-int] backbone[39] (Addgene 199202) via LR gateway cloning, and was inserted into the attP40 landing site using standard phiC31 transgenesis.

### Time series of nuclear morphology and EdU incorporation

To quantify ploidy and nuclear morphology over the course of the fly's life, age matched flies of the genotype *GMR7106-Gal4 / UAS-Stinger; MKRS /+* were collected on eclosion day, and maintained at 25 °C in standard fly vials on standard lab food (approximately 10–15 adult flies per vial), either with or without dried active yeast sprinkled on the food's surface. In a pilot experiment, we observed that *UAS-Stinger* is tightly nuclear-localized in this tissue, whereas other common nuclear-localized fluorophores transgenes (*UAS:GFP[NLS]* [BL4776], *UAS-unc84-GFP* [lab stock] and *UAS:RedStinger* [BL8545]) were not. Flies were flipped onto fresh food every 2 or 3 days. On each day of sampling (3, 7, 14, 21, 28, and 35 days), approximately 6–8 cardia were dissected and stained overnight with Alexa 488-coupled anti-GFP, DAPI, and phalloidin as described above. At each time-point, confocal z-stacks were collected through the entirety of Zone 4 ($n = 5$–6 proventriculus per time point) and analyzed as described below. We also collected samples for eclosion day and for 1 day later, but found that the Gal4 expression pattern extended beyond Zone 4 at these time points, so they were excluded from further analysis.

For *OregonR* aging experiments with EdU, batches of flies were collected on the day that they eclosed and then aged at 25 °C on standard food with yeast powder until the indicated ages. Flies were transferred to EdU+ food 2 days prior to EdU+ food. Anterior-posterior position was calculated from confocal stacks by measuring the shortest distance between the center of each EdU+ and the vertical center of the posterior-most row of cells in Zone 4, hence, EdU+ cells located in the posterior-most row were measured as zero.

### Image quantification

Ploidy was analyzed from DAPI-stained confocal z-stacks, following a protocol modified from ref. 80 and described below, using FIJI software[81]. A subset of the confocal z-stack containing only a single layer of nuclei from the surface of Zone 4 was generated using the "sum intensity" projection method, and region-of-interests were manually drawn for each visible Zone 4 nucleus using *71F06-Gal4 > GFP* signal to ensure that all nuclei measured were in Zone 4. Summed DAPI intensity was calculated using the "integrated density" function. To normalize these values, we used diploid nuclei from Zone 3 as an internal control within each sample. We determined Zone 3 cells to be diploid by direct comparison to diploid ISCs captured under identical imaging parameters. Thus, a diploid standard was calculated for each sample from an averaged value of approximately 15–25 diploid cells within each confocal stack.

Additional nuclear quantification, including cell volume and cell counts were performed with arivis Vision4D software (Zeiss, v4.1.2). GFP+ nuclei were segmented using the Cellpose-based Segmenter, and segmentation was then manually corrected for each sample by splitting or removing nuclei that were erroneously fused and removing artifacts that did not correspond to nuclei. Nuclear volume was extracted for all objects $>10\,\mu m^3$. For cell count, those nuclei missed by Cellpose-based segmentation were manually counted using the "add marker" function while viewing the tissue in 3D. To quantify cell loss during aging, cells that had been shed into the lumen were manually counted in FIJI[81].

As a measure of tissue size for Zone 4, we measured the area of Zone 4 at the widest point of the proventriculus. Each sample value represents the averaged area of two independent measurements of cross-sections of Zone 4, on either side of the esophagus. For area measurements of peritrophic membrane features stained with HPA lectin, WGA, or calcofluor, we made a maximum intensity projection of 3–4 confocal z-slices at the equator of the proventriculus, and measured the area of stained material, averaging two measurements taken for each sample on either side of the esophagus.

### Peritrophic membrane electron microscopy and leakiness assays

"Smurf" assays were performed as previously described[67], as follows. Flies transferred to standard lab food containing 2.5% FD&C Blue #1 dye (Spectrum) by weight, and were examined for gut leakiness after 24 and 48 h. Any flies where blue food was restricted to the proboscis, crop, and gut was scored as wildtype.

Fluorescent bead leakiness analysis were performed similar to previous reports[55,66]. Flies were starved in empty vials for 2 h at 29 °C, then transferred to vials containing half of a standard kimwipe saturated with a 1:50 dilution of Fluoresbrite YG microbeads (diameter 0.056 μm; Polysciences) in 5% sucrose. Flies were dissected between 30–60 min after feeding, and whole guts were fixed and stained with DAPI and phalloidin and/or HPA lectin as described above. Guts were scored under 40X magnification as either intact or containing leaky regions where fluorescent beads were visible outside of the peritrophic membrane.

Electron microscopy was performed in the Harvard Electron Micrscopy Facility. Tissues were prepared as described[69], and longitudinally sectioned anterior midguts were imaged using a Tecnai G2 BioTwin microscope.

### Lifespan and *Pseudomonas* survival experiments

Lifespan experiments were conducted at 29 °C, with 15 mated females per vial on standard fly food supplemented with dried yeast powder, and flipped every 1–2 days. Flies of the genotype *w; tubGal80[ts], UAS:GFP; 71F06-Gal4* were crossed to control or UAS:RNAi lines, and the offspring were reared at 18 °C until eclosion. Adults were collected and mated over a 4-day period at 18 °C, then shifted to 29 °C 3 days later to begin life-span measurements. The number of dead flies was recorded every day.

Infection with *Pseudomonas aeruginosa (PA14)* followed a protocol similar to ref. 69. Five milliliters of LB medium was seeded from a glycerol stock of *PA14* and grown overnight in a shaking 37 °C incubator. The following morning, 1 mL of this culture was diluted in 50 mL of fresh LB and grown for an additional 6 h, shaking at 37 °C. This culture was then diluted 1:3 in 5% sucrose solution. Fly vials containing half of a standard kimwipe, compressed into the bottom of the vial, were loaded with 1.5 mL of this bacterial suspension. In order to promote feeding on the bacteria, flies were starved for 2 h in empty vials immediately prior to bacterial treatment.

These experiments were performed at 29 °C on flies that had been maintained at 29 °C for 15 days prior to *PA14* infection. After a 24 h infection, flies were flipped to fresh vials containing Kimwipes saturated with 5% sucrose solution, and flipped to fresh 5% sucrose solution daily after that. The number of dead flies was recorded approximately every 24 h. Four vials of 8–10 flies each were used per genotype, and the entire experiment was performed in biological duplicate, with similar results in both experiments.

### Graphing and statistical analysis

All graphs were created and statistical tests performed using Prism (GraphPad, v10).

### Reporting summary

Further information on research design is available in the Nature Portfolio Reporting Summary linked to this article.

## Data availability

The snRNA-seq data generated in this study have been deposited in the NCBI Gene Expression Omnibus (GEO) database under accession code GSE301009. The snRNA-seq Atlas is available for data mining at

https://www.flyrnai.org/scRNA/. The raw data used to generate all figures are provided as a Source Data file.

## Code availability
Scripts used to analyze snRNAseq data are provided here: https://github.com/MujeebQadiri/SingleNucleiProventriculusAnalysis/tree/main.

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

## Acknowledgements

We thank Mujeeb Qadiri, Austin Veal, and Aram Comjean for assistance with snRNAseq data analysis and sharing, and Rich Binari for assistance with fly work. We thank Hannah Dayton and Liz Lane for reagents and advice with *Pseudomonas* experiments. We thank our three anonymous Reviewers for proposing several experiments and ideas that improved our study. This work was supported by the National Institute of Health R24 OD031952 awarded to NP. NP is an investigator of the Howard Hughes Medical Institute. This article is subject to HHMI's Open Access to Publications policy. HHMI lab heads have previously granted a non-exclusive CC BY 4.0 license to the public and a sublicensable license to HHMI in their research articles. Pursuant to those licenses, the author-accepted manuscript of this article can be made freely available under a CC BY 4.0 license immediately upon publication.

## Author contributions

B.E.C. conceived of the study, designed and performed all experiments, analyzed data, and wrote the manuscript with supervision from N.P. and input from all authors. S.G.T. and Y.L. assisted with the creation of the snRNAseq atlas, and W.C. and Y.H. performed bioinformatic analysis of the snRNAseq atlas. N.P. supervised the study, provided instrumentation and reagents, and contributed to the writing of the manuscript.

## Competing interests

The authors declare no competing interests.
