## [Peer Review File · Nature Communications]

The *Drosophila* proventriculus lacks stem cells but compensates for age-related cell loss via endoreplication-mediated cell growth

Corresponding Author: Professor Norbert Perrimon

Version 0:

Reviewer comments:

Reviewer #1

(Remarks to the Author)
review of NCOMMS-25-50845

In this manuscript, Ewen-Campen et al. re-investigated the proventriculus zone 4 cells, which were previously proposed to contain multipotent stem cells by Singh, Zeng, Zheng, and Hou. In contrast to the previous publication, the authors, through genetic and lineage analysis, found that the proventriculus zone 4 cells do not contain stem cells because they have no mitotic activity, do not generate clonal lineages, and do not respond to tissue damage. However, to put this in perspective, Sing et al.'s 2008 claim of PV4 stem cells was similar to a claim by Takashima et al. in 2008 for the hindgut. However, in 2009, Fox and Spradling 2009, disproved the claim and showed it regulated from defects in the lineage system. The field realized that Sing et al.'s data was similar but even lower quality than Takashima et al., and consequently this claim has not been believed for at least 15 years. There the paper needs more than just to correct the record on this old claim.

The authors also reported that the proventriculus zone 4 cells undergo continuous endoreplication in a nutrient-dependent manner. They concluded PV4 cells are lost as the fly ages, and claim this results in functional defects in the peritrophic membrane (PM) and an increased susceptibility to oral bacterial infections. There is a substantial literature on the early aging of the *Drosophila* gut, at least under lab conditions. These are potentially interesting findings, but they need to be better supported experimentally in ways that add insight. Currently the novelty of the findings is shallow.

The descriptive data regarding proventriculus zone 4 cells number under homeostasis as well as in various nutrient and stress conditions, needs to be better supported with more data. What is the rate of polyploidization of PV4 and how does it vary with nutrition? Do these changes affect the number of PV4 cells relative to other cells contributing to PM? Do you have information on what it responds to? Insulin levels, or particular nutrients. Does the PV4 response change PM structure over time?

The experiments that utilize the tumor cachexia model contain a systematic flaw due to the use of *esg-GAL4* to manipulate the proventriculus with the assumption that these effects act only through changes in nutrition via action on ISCs. It was demonstrated (ref 25) that *esg-GAL4* is highly active in the corpus allatum (CA), the major source of adult juvenile hormone. JH is a major regulator of reproductive diapause during which its level is seen to decline sharply and possibly in circadian fluctuations in gut activity. JH might well effect PV cells since many express JH binding proteins as well.

Below are more detailed point-by-point comments.

1. Figure 1F, zone 4 cells in the 5 days EdU image seem disorganized or "lose a clearly monolayered epithelial structure" as defined later in the text (Line 397). Is this representative? Given that the authors later reported an age-related phenotype, please clarify how the fly age is controlled for each experiment within the study.
2. Please elaborate on the "observed a marked decrease in EdU+ cells in Zone 4, suggesting that the EdU signal was diluted by continued cycling" (Line 127)? If the proventriculus zone 4 cells do not have mitotic activity as the authors

suggested later, would an accumulation of the EdU signal from the EdU exposure period (3 days in the case of Figure S1B) be expected?

3. The authors observed (Line 133) that “after 1d of EdU feeding, Edu+ nuclei tended to be small in volume and located at the posterior limit of Zone 4 whereas after 5d of feeding labeled nuclei were larger and could be seen a few cell diameters anteriorly.” Please comment on whether this is indicative of the zone 4 cell migration.

4. The authors concluded that the proventriculus zone 4 cells are sensitive to systemic nutrient status (Line 147). Under physiological conditions, nutrient status fluctuates over time, especially for animals like wild flies. How would the proventriculus zone 4 cells respond under physiological conditions, e.g. when switching between rich and poor nutrient status? Also, how are these signals transmitted to the proventriculus zone 4 cells? Do you find expression of any receptors or regulatory molecules responsible for nutrient status (e.g. insulin receptor) in these cells?

5. Line 147, do you mean Figure 1H instead of Figure 1E?

6. The manuscript can be written more concisely with a clearer focus on re-examining proventriculus zone 4 cells. The authors reported an interesting nutrient-dependent endoreplication phenomenon in the proventriculus zone 4 cells (Figure 1G&H) at the beginning, but quickly shifted gears and used a significant length of writing reporting major cell types, transcriptomes, and associated genetic lines for the proventriculus (Line 167) before coming back on examining the cell biology of proventriculus zone 4 cells. Much of this information is not directly relevant to the focus of the study, and some of the languages used in the manuscript have actually been learned by several previous studies in great depth. It would be more succinct to highlight findings that have not been reported or thoroughly examined in the past, such as zone 3-4 junctional cells, and reference what was known in the appropriate literature.

7. Do you find any PM defects in *esg-Gal4, tubGal80ts > UAS:yki3SA*, and *esg-Gal4* and *dMef2-Gal4* driving *ImpL2* and/or *upd3* overexpression flies, where proventriculus zone 4 cells have slower endoreplication (Line 150-160)?

8. Please provide quantifications for Figure 3A.

9. In Figure 7D, zone 4 cells decrease by ~21 cells by 14d; however, Figure 7F only reports ~1 cell loss into the lumen by this time. Please clarify these quantifications.

10. Line 394-395, do you mean Figure S7B instead of Figure S6B? Line 398, do you mean Figure S7C instead of Figure S6C?

11. Do you find that the loss of proventriculus zone 4 cells correlates with any other cell biological characteristics of the cells?

12. Line 411, do you mean Figure S8 instead of Figure S7?

13. The authors reported alterations in the PM structure upon perturbations to endocycling in the proventriculus zone 4 cells (Line 430-450 and Figure 8). PM is a dynamic structure and varies by animal and tissue location. More precise and quantitative approaches would be necessary to conclude the PM phenotype. For example, do you find changes in key PM-producing genes upon your treatments (e.g. using qRT-PCR or RNAseq)? Also, the PM is composed of multiple layers. Could you examine the detailed PM structure using electron microscopy? Given that the PM is largely produced in the proventriculus, but its main function occurs in the midgut, it would be informative to provide more characterizations of the PM defects in both the proventriculus and midgut in a layer-specific manner.

14. Figure 8F, do the “Increased PM” conditions in Figure 8A show a better fly survival post-pathogen infection?

15. Line 910, I am not sure what “single-cell epithelium” means.

16. The authors conducted scRNAseq on well-fed flies and flies with reduced systemic insulin signal (Line 575). Did the authors compare the data from the two conditions? Where was the analysis for the cachexic flies? What datasets were used for the atlas analysis?

17. In general, the figure legends need to be reorganized. Information regarding scale bar size and experimental replication needs to be precisely summarized. Currently, this information is flowing around and sometimes missing. Some of the experiments that led to the main conclusions in the paper need more replication.

18. Please provide more detailed descriptions of statistical analysis in the materials and methods. In many places when t-tests were used, it was unclear whether the data passed the normality test.

19. Please add languages to the discussion on how this study differs from the previous study by Singh, Zeng, Zheng, and Hou that led to the opposite conclusions.

Reviewer #2

(Remarks to the Author)

Reviewer #3

(Remarks to the Author)

Summary:

In this study, the authors reexamine an anterior region of the fruit fly intestine, known as the proventriculus, which was reported to be maintained by multipotent stem cells. They definitively show, using cell cycle markers and lineage tracing techniques, that stem cells do not exist in this tissue, correcting the initial report. Using a variety of genetic and genomic techniques the cell-types within proventriculus are clearly defined providing a new toolkit of Gal4 drivers for future studies. In addition, the authors discover that cells that cycle (S-phase+) in zone 4 are endocycling, instead of mitotically dividing. Endoreplication appears to be age and nutrition dependent, which may be functionally necessary for the synthesis of the peritrophic member. Secretion of the peritrophic membrane is essential to protect most insect intestines from damage upon ingestion of food as well as pathogenic microorganisms. This is further demonstrated by enhanced sensitivity to P.

aeruginosa in flies with the insulin receptor knocked down in zone 4 cells of the proventriculus. Overall, this study provides a comprehensive analysis of the proventriculus cellular organization and gene expression program. However, many of the conclusions, particularly in regard to polyploidy are not supported by the experiments performed (i.e. nuclear volume, instead of DNA content measurements). Therefore, the following major and minor comments are strongly recommended to be addressed:

Major Comments:

1. Nuclear volume does not equal nuclear ploidy and cannot be used as a proxy for concluding that a change has occurred. If the authors want to conclude that ploidy is changing with nutrition and age then it needs to be measured. Ploidy can be measured using confocal imaging of DAPI or Hoescht stained nuclei using a diploid cell type in the guts as an internal control (i.e. intestinal stem cells or enteroendocrine cells). Therefore, existing images acquired in this study could be used and reanalyzed to draw these conclusions.
2. The ploidy measurements would be helpful to further understand the organization of EdU+ nuclei as it was noted that at 1d and 5d post EdU feeding the nuclei were localized in different regions of zone 4. Spatial ploidy, instead of nuclear volume, analysis is necessary to conclude that ploidy changes with age and nutrition in this region of the tissue.
3. The authors meticulously and rigorously identify as well as generate cell-type specific Gal4 drivers in this study yet they are under-utilized. An example of this is in regard to the genetic ablation experiment in Figure 5. Cell death may need to be induced in zone 4 itself to activate the endocycle. That is the case for other fly tissues that induce polyploidy in response to injury/damage. Gal4 drivers specific to zones 2/3, 3/4, and/or 4 should be used to ablate cells and test if zone 4 cells are truly refractory to enhanced endoreplication following injury. In addition, the DEMISE system could be used to couple apoptosis with genetic manipulations (RNAi or overexpression) of other target genes (Cohen et al. 2018).
4. This study concludes that the proventriculus response to damage is unique from other models of wound- or damage-induced polyploidy yet the assays used are not analogous to prior studies (local vs distal damage- comment #3) and the zone 4 cells appear to be polyploid (Figure S6). The authors may not see a response to X-rays, chemicals, or pathogens as polyploid cells are known to be more resilient to stress. Do the treatments (Figure S5) lead to local damage in zone 4? The high ploidy of zone 4 cells (4C to 40C) may inhibit further polyploidization, a distinction from the epithelium and hindgut pyloric cells which are diploid (2C) prior to injury.
5. One of the most impactful parts of this study is the potential physiological role for endoreplication in the synthesis of the peritrophic membrane. However, the study lacks a direct demonstration of this conclusion. To do so, endoreplication in zone 4 cells should be directly inhibited using cycE RNAi or fzf RNAi, instead of knockdown of the insulin receptor. Additionally, it is unclear when in development the peritrophic membrane is produced or if it has to be continually maintained? If it has to be continuously synthesized during adult life than this may explain why the zone 4 cells continue to endocycle with age.
6. The authors could use the dextran permeability assay (Kuraishi et al. PNAS 2011) to determine if the peritrophic membrane is compromised when endoreplication is upregulated or inhibited. It is unclear if the changes in lectin staining are indicative of a functional change.

Minor Comments:

1. The authors should consider moving Figure 6 (GTRACE) to the supplement as it is redundant with Figure 4 (MARCM).
2. Line 333-335: Liver hepatocytes are derived from the endoderm and well-documented to endoreplicate in response to tissue damage, so there is insufficient evidence to make a claim of an ectoderm vs endoderm difference in endoreplication.
3. Miss cited figures (i.e. Fig S6C – line 397-8 does not exist). Figure S7 legend is missing a description for part (C).
4. Several graphs (i.e. Figure 1H and 6F) are missing significance and the statistical test if used it is not reported.
5. Check image stains are labeled in all figures (i.e. Figure 2E is missing this info)
6. It would be helpful to provide the guts (n) and PH3+ nuclei (#) for intestines in Figure 3B as well.

Reviewer #4

(Remarks to the Author)

In this thorough and carefully conducted study, the authors investigate the mechanism by which tissue homeostasis is maintained in the *Drosophila* adult proventriculus. A previous study reported on the presence of BrdU+ cells in the proventriculus and proposed that it therefore contains mitotically active stem cells. Here the authors reexamine this conclusion and provide convincing evidence that these cells are endocycling rather than undergoing mitosis and therefore are not stem cells. They reach this conclusion through a combination of approaches, including stains for mitotic markers and clonal analysis in a wide range of experimental conditions designed to enrich for mitotic cells if they exist in this tissue. However, despite this extensive search, they do not find evidence of cells in mitosis. Suitable positive controls are included to confirm that their methods are working as expected. They also generate a single cell atlas of the proventriculus and use the results, along with a screen and a recently developed split-Gal4 tool to specifically target the region of the tissue where cells in S-phase can be found. Interestingly, they find that cells in this region are polyploid and that the ploidy increases with age, suggesting that they continue to endocycle during adulthood. The authors propose that this may be a mechanism to compensate for age-dependent cell loss, which they provide evidence for. Overall, I think this study will be a valuable contribution to the field and I am supportive of publication. I have only a few minor comments, which I recommend the authors consider before publication:

1. I am confused by the results described in the paragraph from lines 132-139 (Figure 1D is referenced but I think the authors are referring to Figure 1F): if the cells are continuously endocycling, why would the size of EdU+ nuclei increase over 5 days? Wouldn't there be a distribution of nuclear sizes at any given time point when EdU is administered such that both large and small nuclei could be labeled even at day 1?

2. On line 147, it seems that Figure 1H should be referenced (not Figure 1E)
3. On Line 252-253, the authors state that they never saw cells travel from Zone 4 to any other portion of the proventriculus but, without live imaging or clonal analysis, how is it possible to track cell movement?
4. The data in Fig 4D show that there are, in fact, a small number of large clones in the proventriculus at 14 days after heat shock (dark grey bar at the top of the third column from the left) but this is not discussed in the text. How did these larger clones arise if the cells cannot undergo mitosis?
5. The code used to analyze the scRNAseq data as well as the final Seurat objects should be made available.

Version 1:

Reviewer comments:

Reviewer #1

(Remarks to the Author)

The changes made in the revised version have addressed the major concerns raised in the original review in a satisfactory manner. I now favor publication without further substantive changes.

The paper is framed as a re-investigation of a 2011 report describing proventriculus (PV) stem cells that were never confirmed and fell into obscurity. However, this paper goes beyond classical stem cells and provides convincing evidence of a different and very interesting alternative phenomenon related to "wound-induced polyploidy," an alternative repair system based on induced endocycles rather than mitotic cycles. Midgut and hindgut enterocytes, and many other cells are known under some conditions to polyploidize in response to tissue damage, as has now been well studied previously by multiple groups including the authors. Here they show that PV4 cells undergo scattered endocycling in a ring-like pattern during much of adult life, in the absence of a known source of tissue damage.

An interesting potential explanation would be that active peritrophic membrane (PM) production is intrinsically damaging to the posterior rim of PV4 cells. The cells may be responding to cell loss with induced polyploidization of some remaining nearby PV4 cells to maintain the functional capacity of this region, which is thought to play a critical role by generating the major chitin-rich inner layer of the peritrophic membrane. This layer is involved in digestion, and restricting access of gut microbiota to enterocyte tissue. These findings recall the well known theme in studies of the gut that the normal tasks of some cells are intrinsically damaging. This is the generally accepted reason for the rapid turnover of midgut enterocytes and the presence of highly active stem cells to supply replacements. Damaging food and microorganisms are thought to be largely excluded from the upper PV, and this is a rationale for the absence of active stem cells. Something must be different in this region of PV4, and the capacity of the endocycle-based repair may eventually be inadequate and contribute to an age and metabolism-related loss of homeostasis and normal tissue structure. A similar scenario of polyploidization incompletely counteracting damage leading to slow progression of disease was described in the mouse model of Fuchs Endothelial Corneal Dystrophy analyzed in reference 14. Endocycling in the pattern described might serve as a marker for such early aging.

The authors in the revised version make a strong case for cell turnover and compensatory polyploidization (rather than cell generation) in PV4. They now show that the average ploidy as well as nuclear volume of zone 4 cells increases over the lifespan. They do not currently propose a reason for this behavior other than to call it a "unique pattern of endoreplication." I think the paper would be further improved by a little more discussion of what factors might lead to such a unique pattern of cell cycle behavior.

Reviewer #3

(Remarks to the Author)

Summary:

This revised manuscript addresses the major and minor concerns of the original study. The ploidy analysis now proves that zone 4 cells in the proventriculus endocycle with age and appear to do so to compensate for cell loss. The authors find that cell loss occurs naturally with age as well as in response to genetically induced-apoptosis. This is consistent with other studies performed to date in fruit fly. This study also provides a comprehensive analysis of the proventriculus cellular organization, its gene expression program, as well as identification of a novel Gal4 driver to regulate gene expression in zone 4. The study debunks previous report of stem cells in the tissue and finds that tissue homeostasis appears to be maintained by cell growth, instead of cell division. Despite the many different genetic manipulations performed, the authors were not able to find a condition to directly inhibit endoreplication in zone 4 cells. Therefore, the physiological importance of endoreplication to the intestine's peritrophic membrane synthesis as well as infection resistance remain inconclusive lessening the impact of this study.

Remaining Comments:

1. A major conclusion of this paper is that endoreplication with age or injury compensates for cell loss. The authors should be able to definitively show this by summing the nuclear ploidy in young vs old or uninjured vs repaired tissues. If ploidy is compensating for cell loss, then the total ploidy of the two conditions will remain unchanged despite a reduction in cell number.
2. The authors still claim that their findings are unique (line 625-635) from other studies on injury or age-induced polyploidy.

However, the uniqueness is subtle at best as Nandakumar et al. 2020 demonstrated that neuron ploidy increases with age via endocycle. As for nutrient dependance, endoreplication is well-characterized to be sensitive to nutritional inputs. Therefore, the changes observed are more likely tissue intrinsic and due to cell type specific differences in longevity/turnover rate.

Reviewer #4

(Remarks to the Author)

With this revision, the authors have fully addressed my concerns.

REVIEWER COMMENTS

Reviewer #1 (Remarks to the Author):

In this manuscript, Ewen-Campen et al. re-investigated the proventriculus zone 4 cells, which were previously proposed to contain multipotent stem cells by Singh, Zeng, Zheng, and Hou. In contrast to the previous publication, the authors, through genetic and lineage analysis, found that the proventriculus zone 4 cells do not contain stem cells because they have no mitotic activity, do not generate clonal lineages, and do not respond to tissue damage. However, to put this in perspective, Sing et al.'s 2008 claim of PV4 stem cells was similar to a claim by Takashima et al. in 2008 for the hindgut. However, in 2009, Fox and Spradling 2009, disproved the claim and showed it regulated from defects in the lineage system. The field realized that Sing et al.'s data was similar but even lower quality than Takashima et al., and consequently this claim has not been believed for at least 15 years. Therefore, the paper needs more than just to correct the record on this old claim.

We thank the reviewers for this feedback, and respectfully disagree that there is clarity or agreement in the literature about whether or not the proventriculus contains stem cells.

While the reviewers are correct that Fox & Spradling (2009) disproved the existence of stem cells in the hindgut, it does not follow that one can draw any conclusions about the proventriculus, a different tissue. And in fact, as we show in this manuscript, the cycling dynamics of these two gut regions differ: the hindgut only becomes polyploid in response to tissue injury, whereas in the proventriculus there is continual endocycling even in the absence of any exogenous damage.

Second, while it may be true that some researchers in the field viewed Singh *et al.* (2011) as unpersuasive, there have been zero publications re-addressing this issue to date. Clarifying this issue of whether or not stem cells are present in this tissue is a primary motivation for our manuscript, and has not been addressed in the literature to date.

The authors also reported that the proventriculus zone 4 cells undergo continuous endoreplication in a nutrient-dependent manner. They concluded PV4 cells are lost as the fly ages, and claim this results in functional defects in the peritrophic membrane (PM) and an increased susceptibility to oral bacterial infections. There is a substantial literature on the early aging of the *Drosophila* gut, at least under lab conditions. These are potentially interesting findings, but they need to be better supported experimentally in ways that add insight. Currently the novelty of the findings is shallow.

In our revised manuscript we have deepened our analysis of the peritrophic membrane by performing multiple additional assays and quantification to describe the PM phenotype, now

shown in revised Figure 8 and Figure 9, as well as Supplemental Figure S11. In our initial submission, our analysis was limited to HPA-Lectin staining; in our revised manuscript we have added stainings with calcofluor, which stains chitin, as well as wheat germ agglutinin, which stains a distinct set of glycosylated targets. We have also now quantified the amount of stained PM material in the proventriculus, and showed consistent changes in all three labels following genetic manipulations to the endocycle, as would be predicted by a direct relationship between metabolic output of these cells and their ploidy state/cell size. As described below, we also examine several aspects of PM function in the midgut as well (using TEM, smurf assay, and a fluorescent bead retention assay.)

The descriptive data regarding proventriculus zone 4 cells number under homeostasis as well as in various nutrient and stress conditions, needs to be better supported with more data. What is the rate of polyploidization of PV4 and how does it vary with nutrition? Do these changes affect the number of PV4 cells relative to other cells contributing to PM? Do you have information on what it responds? Insulin levels, or particular nutrients. Does the PV4 response change PM structure over time?

We have re-analyzed our time-course analysis of nuclear changes in Zone 4 in order to directly measure ploidy (rather than nuclear volume, as we had previously done.) This allowed us to directly show changes in ploidy over the first 35 days of the fly's life, shown in a revised Figure 6 and described in the text. We show that both ploidy and cell number are influenced by yeast supplementation (Figure 6C,D), and we also now directly show that endocycling can be induced by cell loss from the tissue. Our RNAi experiments with *InR* (revised Figure 9) demonstrate that Zone 4 cells respond to nutritional signals in a cell autonomous fashion, which we also clarify in the text of the manuscript.

The experiments that utilize the tumor cachexia model contain a systematic flaw due to the use of *esg-GAL4* to manipulate the proventriculus with the assumption that these effects act only through changes in nutrition via action on ISCs. It was demonstrated (ref 25) that *esg-GAL4* is highly active in the corpus allatum (CA), the major source of adult juvenile hormone. JH is a major regulator of reproductive diapause during which its level is seen to decline sharply and possibly in circadian fluctuations in gut activity. JH might well effect PV cells since many express JH binding proteins as well.

We thank the authors for pointing out this caveat of our *esg-Gal4* experiments, and have noted this in the text. We have now emphasized in the text that we also over-expressed *Impl2* and *Upd3* from muscles (*dMef2-Gal4*) also drives a significant decrease in cell cycling in the proventriculus, which provides independent support to the conclusion that systemic nutrition is influencing the cycling status of these cells.

Below are more detailed point-by-point comments.

1. Figure 1F, zone 4 cells in the 5 days EdU image seem disorganized or “lose a clearly monolayered epithelial structure” as defined later in the text (Line 397). Is this

representative? Given that the authors later reported an age-related phenotype, please clarify how the fly age is controlled for each experiment within the study.

Based on helpful comments from all three reviewers, we have now removed the data shown in Figure 1F (comparing 1d EdU vs. 5d EdU feeding), as we agree no real conclusions can be drawn from this data.

We have also clarified in the Methods section that, except as noted in the text, all experiments show flies under 7 days post-eclosion.

2. Please elaborate on the “observed a marked decrease in EdU+ cells in Zone 4, suggesting that the EdU signal was diluted by continued cycling” (Line 127)? If the proventriculus zone 4 cells do not have mitotic activity as the authors suggested later, would an accumulation of the EdU signal from the EdU exposure period (3 days in the case of Figure S1B) be expected?

We have clarified the text to explain that continual endocycling dilutes the EdU as the amount of EdU-labeled DNA decreases as a proportion of the total DNA in the nucleus, similar to what we and others have observed in midgut ECs, which endocycle.

3. The authors observed (Line 133) that “after 1d of EdU feeding, Edu+ nuclei tended to be small in volume and located at the posterior limit of Zone 4 whereas after 5d of feeding labeled nuclei were larger and could be seen a few cell diameters anteriorly.” Please comment on whether this is indicative of the zone 4 cell migration.

Please see above, in our response to (1).

4. The authors concluded that the proventriculus zone 4 cells are sensitive to systemic nutrient status (Line 147). Under physiological conditions, nutrient status fluctuates over time, especially for animals like wild flies. How would the proventriculus zone 4 cells respond under physiological conditions, e.g. when switching between rich and poor nutrient status? Also, how are these signals transmitted to the proventriculus zone 4 cells? Do you find expression of any receptors or regulatory molecules responsible for nutrient status (e.g. insulin receptor) in these cells?

Our experiments with InR RNAi suggest that Zone 4 cells respond cell autonomously to insulin signaling, as knocking down the insulin receptor in these cells abolishes endocycling. We agree that in future studies it will be very interesting to characterize how major fluctuations in food availability are mechanistically linked to the peritrophic membrane machinery.

5. Line 147, do you mean Figure 1H instead of Figure 1E?

Fixed.

6. The manuscript can be written more concisely with a clearer focus on re-examining proventriculus zone 4 cells. The authors reported an interesting nutrient-dependent endoreplication phenomenon in the proventriculus zone 4 cells (Figure 1G&H) at the beginning, but quickly shifted gears and used a significant length of writing reporting major cell types, transcriptomes, and associated genetic lines for the proventriculus

(Line 167) before coming back on examining the cell biology of proventriculus zone 4 cells. Much of this information is not directly relevant to the focus of the study, and some of the languages used in the manuscript have actually been learned by several previous studies in great depth. It would be more succinct to highlight findings that have not been reported or thoroughly examined in the past, such as zone 3-4 junctional cells, and reference what was known in the appropriate literature.

We agree with the Reviewer #1 that Zhu *et al.*'s 2024 PNAS paper established an immensely valuable single cell atlas of the proventriculus, which has provided many insights into the cell types and biology of this understudied tissue. We ultimately decided to include our own snRNAseq section for three reasons 1) because we wound up using different cell-type specific Gal4 lines than those described by Zhu *et al.* and we wanted to explain their origins; 2) because our atlas was generated independently yet was highly concordant, we feel that it serves to strengthen the conclusion that the major cell types of this tissue have indeed now been identified; 3) because we mention the lack of markers for stem cells, we wished to explain in some detail how we had generated our atlas.

That said, if the manuscript Editor agrees that this section is excessively detailed, we can easily shift parts of it to a supplement.

7. Do you find any PM defects in *esg-Gal4*, *tubGal80ts > UAS:yki3SA*, and *esg-Gal4* and *dMef2-Gal4* driving *ImpL2* and/or *upd3* overexpression flies, where proventriculus zone 4 cells have slower endoreplication (Line 150-160)?

We attempted to analyze the PM in Yki gut tumor flies, but found that these tissues are extraordinarily malformed and disrupted by the presence of massive tumors, which makes this analysis exceedingly difficult to perform.

8. Please provide quantifications for Figure 3A.

We have quantified these data, including # of EdU cells, ploidy, and tissue size, in a revised Supplemental Figure 4.

9. In Figure 7D, zone 4 cells decrease by ~21 cells by 14d; however, Figure 7F only reports ~1 cell loss into the lumen by this time. Please clarify these quantifications.

We have clarified in the text that the images and counts of cell loss shown in 7E-7F (now Figure 6E-6F in the revised paper) are "snap-shots" of cells being actively shed at the time of dissection and staining, whereas the total cell counts are the true and complete number of cells remaining in Zone 4 and thus account for the full time period.

10. Line 394-395, do you mean Figure S7B instead of Figure S6B? Line 398, do you mean Figure S7C instead of Figure S6C?

These and other Figure references have been corrected.

11. Do you find that the loss of proventriculus zone 4 cells correlates with any other cell biological characteristics of the cells?

We have added a note in the text to state that we do not observe any obvious morphological changes in the cells being lost. We also note here for the reviewer that we are now researching the molecular factors involved in cell loss from the proventriculus for a future manuscript.

12. Line 411, do you mean Figure S8 instead of Figure S7?

Fixed.

13. The authors reported alterations in the PM structure upon perturbations to endocycling in the proventriculus zone 4 cells (Line 430-450 and Figure 8). PM is a dynamic structure and varies by animal and tissue location. More precise and quantitative approaches would be necessary to conclude the PM phenotype. For example, do you find changes in key PM-producing genes upon your treatments (e.g. using qRT-PCR or RNAseq)? Also, the PM is composed of multiple layers. Could you examine the detailed PM structure using electron microscopy? Given that the PM is largely produced in the proventriculus, but its main function occurs in the midgut, it would be informative to provide more characterizations of the PM defects in both the proventriculus and midgut in a layer-specific manner.

We thank the Reviewer for these helpful suggestions, and as the Reviewer correctly notes, the PM is a highly dynamic structure that varies between individuals and over time, and thus any description of a PM phenotype indeed requires quantification over multiple individuals.

We have now added a number of additional characterizations and quantifications of the peritrophic membrane phenotypes associated with increased and decreased endocycling. These are included in revised Figure 8 and Figure 9, as well as new Figure S11.

Specifically, we now characterize the increase and decreased peritrophic membrane synthesis using three independent dyes: calcofluor (which binds chitin, a major component of one of the four PM layers), and the two lectins HPA-lectin and wheat germ agglutinin, which have distinct binding affinities for glycosylated molecules. We have quantified the amount of PM using each of these three stains by measuring the area of PM adjacent to Zone 4 at the widest point of the peritrophic membrane, representing a snapshot of the amount of PM material being secreted by these cells. These results all show significant support for the hypothesis that increasing or decreasing endocycling in Zone 4 leads to lower and higher levels of PM secretion, respectively.

We also performed three additional assays for PM structure and function in the midgut: we examined the PM via electron microscopy, we performed the “Smurf assay” for gut leakiness, and we performed a fluorescent bead retention assay (which was also suggested by other Reviewers.) These results are shown in Figure S11, and none of these assays revealed consistent or significant differences between control and *InR-RNAi* flies, which suggests that, under standard rearing conditions and in the absence of infection, the peritrophic membrane retains basic functionality and structure even when synthesis is reduced in Zone 4 by blocking endocycling. However, given the results with *Pe* infection, there must be some compromised activity of the PM in terms of infection resistance, suggesting that endocycling in this system confers robustness to the digestive system in the face of environmental challenges.

We also note that we made multiple attempts to perform qPCR for *chs2* to address the Reviewer's comment, but unfortunately found that technical challenges made these experiments excessively difficult for this manuscript. Specifically, we found that to obtain sufficient total RNA from these tissue would require $\sim \geq 100$ dissected proventriculi for each sample, across three replicates and four genotypes, all of which would need to be aged to 14-21d at 29°C (i.e. 1,200 dissections.) Ultimately, we believe that a directly demonstration of varying chitin levels (calcofluor staining), which is the functional product of *Chs2*, is more direct than the demonstration that transcription levels vary with ploidy, which is well established in the literature as a general principle.

14. Figure 8F, do the “Increased PM” conditions in Figure 8A show a better fly survival post-pathogen infection?

We decided against performing a post-infection survival experiment in flies with increased endocycling (UAS-cycE, UAS-myc) because this manipulation is not physiologically realistic (whereas disrupting endocycling serves as a straightforward loss-of-function analysis), and therefore unlikely to meaningfully add to the conclusions of our study. We believe that the PM assays we perform directly support the hypothesis that endocycling levels correlate with synthetic and secretory activity of these cells. In contrast, it is not clear whether we would expect a non-physiologically high PM secretion to confer additional infection resistance.

15. Line 910, I am not sure what “single-cell epithelium” means.

This typo has been corrected (now reads “simple epithelium”)

16. The authors conducted scRNAseq on well-fed flies and flies with reduced systemic insulin signal (Line 575). Did the authors compare the data from the two conditions? Where was the analysis for the cachexic flies? What datasets were used for the atlas analysis?

Our snRNAseq atlas contains all nuclei collected from both control and cachectic flies (see Methods), and in this manuscript we do not present bioinformatic analyses of differences between the two datasets as these were not relevant to the present study. Additional analyses of differences between these datasets will likely be the subject of a future manuscript.

17. In general, the figure legends need to be reorganized. Information regarding scale bar size and experimental replication needs to be precisely summarized. Currently, this information is flowing around and sometimes missing. Some of the experiments that led to the main conclusions in the paper need more replication.

We have endeavored to fix this issue in all instances, and have added scale bar and replication information in those instances where it was lacking.

18. Please provide more detailed descriptions of statistical analysis in the materials and methods. In many places when t-tests were used, it was unclear whether the data passed the normality test.

We have added additional information about all of the statistical tests performed, and have mentioned those cases where adjustments were used for non-normally distributed data.

19. Please add languages to the discussion on how this study differs from the previous study by Singh, Zeng, Zheng, and Hou that led to the opposite conclusions.

We have added a section to the first paragraph of the Discussion addressing this.

Reviewer #2 (Remarks to the Author):

We thank Reviewers #1 and #2 for their helpful feedback.

Reviewer #3 (Remarks to the Author):

Summary:

In this study, the authors reexamine an anterior region of the fruit fly intestine, known as the proventriculus, which was reported to be maintained by multipotent stem cells. They definitively show, using cell cycle markers and lineage tracing techniques, that stem cells do not exist in this tissue, correcting the initial report. Using a variety of genetic and genomic techniques the cell-types within proventriculus are clearly defined providing a new toolkit of Gal4 drivers for future studies. In addition, the authors discover that cells that cycle (S-phase+) in zone 4 are endocycling, instead of mitotically dividing.

Endoreplication appears to be age and nutrition dependent, which may be functionally necessary for the synthesis of the peritrophic membrane. Secretion of the peritrophic membrane is essential to protect most insect intestines from damage upon ingestion of food as well as pathogenic microorganisms. This is further demonstrated by enhanced sensitivity to *P. aeruginosa* in flies with the insulin receptor knocked down in zone 4 cells of the proventriculus.

Overall, this study provides a comprehensive analysis of the proventriculus cellular organization and gene expression program. However, many of the conclusions, particularly in regard to polyploidy are not supported by the experiments performed (i.e. nuclear volume, instead of DNA content measurements). Therefore, the following major and minor comments are strongly recommended to be addressed:

Major Comments:

1. Nuclear volume does not equal nuclear ploidy and cannot be used as a proxy for concluding that a change has occurred. If the authors want to conclude that ploidy is changing with nutrition and age then it needs to be measured. Ploidy can be measured using confocal imaging of DAPI or Hoescht stained nuclei using a diploid cell type in the guts as an internal control (i.e. intestinal stem cells or enteroendocrine cells). Therefore, existing images acquired in this study could be used and reanalyzed to draw these conclusions.

We are very grateful to the Reviewer for this feedback, and have re-analyzed every relevant dataset using a direct measure of ploidy instead of nuclear volume. We validated that the nuclei directly posterior to Zone 4 (a region called Zone 3) are diploid and thus used internal ploidy controls within each confocal image that we captured. While these re-analyses did not change the overall conclusions of the paper, the use of a direct assay for ploidy is obviously preferable and more informative.

2. The ploidy measurements would be helpful to further understand the organization of EdU+ nuclei as it was noted that at 1d and 5d post EdU feeding the nuclei were localized in different regions of zone 4. Spatial ploidy, instead of nuclear volume, analysis is necessary to conclude that ploidy changes with age and nutrition in this region of the tissue.

As mentioned above, all three Reviewers identified issues with drawing any conclusions from Figure 1F, and we have thus removed this from the revised manuscript. This removal does not change any of our conclusions.

3. The authors meticulously and rigorously identify as well as generate cell-type specific Gal4 drivers in this study yet they are under-utilized. An example of this is in regard to the genetic ablation experiment in Figure 5. Cell death may need to be induced in zone 4 itself to activate the endocycle. That is the case for other fly tissues that induce polyploidy in response to injury/damage. Gal4 drivers specific to zones 2/3, 3/4, and/or 4 should be used to ablate cells and test if zone 4 cells are truly refractory to enhanced endoreplication following injury. In addition, the DEMISE system could be used to couple apoptosis with genetic manipulations (RNAi or overexpression) of other target genes (Cohen et al. 2018).

We are grateful for this suggestion, and have now performed genetic ablation experiments in Zone 4 itself, shown in a new Figure 7. These experiments led to an important new observation, which is that cell death within Zone 4 itself does result in increased endocycling. This result is an important improvement to our paper because it directly supports the hypothesis that endocycling is a compensatory mechanism to account for cell loss from the tissue.

We had initially focused on genetic ablation in other regions of the proventriculus specifically to address previous claims from (Singh et al 2011) who claimed that proventriculus stem cells divide in order to replace dying cells elsewhere in the tissue, a claim that our data refutes.

However, the Reviewer's suggestion to extend these to Zone 4 itself was very informative, as is the suggestion to use DEMISE in future experiments to understand the mechanism by which Zone 4 endocycles in response to damage.

Relatedly, given the observation that endocycling both (1) occurs in the absence of exogenous damage but (2) can be induced by damage to the tissue, we used an aging experiment to reconcile these two facts. Specifically, we find that when cell loss ramps up in older flies, there is a distinct change in the spatial distribution of endocycling cells, which we propose is a response to cell loss in aging flies. This is described in the text and in Figure 7E-F.

4. This study concludes that the proventriculus response to damage is unique from other models of wound- or damage-induced polyploidy yet the assays used are not analogous to prior studies (local vs distal damage- comment #3) and the zone 4 cells appear to be polyploid (Figure S6). The authors may not see a response to X-rays, chemicals, or pathogens as polyploid cells are known to be more resilient to stress. Do the treatments (Figure S5) lead to local damage in zone 4? The high ploidy of zone 4 cells (4C to 40C) may inhibit further polyploidization, a distinction from the epithelium and hindgut pyloric cells which are diploid (2C) prior to injury.

See response to #3 - we agree that direct damage to Zone 4 was an important addition to the manuscript.

The reason that we chose to include the X-ray, bleomycin, ECC15 experiments in S5 despite the caveats noted by the Reviewer is due to the inherent difficulty in proving a "negative" result - in this case, that there really are no MARCM clones generated from Zone 4. We believe that the more evidence we can provide that, under a wide range of experimental manipulations, the stronger our case is that previous observations were wrong and that our results are robust.

5. One of the most impactful parts of this study is the potential physiological role for endoreplication in the synthesis of the peritrophic membrane. However, the study lacks a direct demonstration of this conclusion. To do so, endoreplication in zone 4 cells should be directly inhibited using *cycE* RNAi or *fzr* RNAi, instead of knockdown of the insulin receptor. Additionally, it is unclear when in development the peritrophic membrane is produced or if it has to be continually maintained? If it has to be continuously synthesized during adult life than this may explain why the zone 4 cells continue to endocycle with age.

We have now performed RNAi experiments using four additional candidates for blocking the endocycle: *fzr*, *cycE*, *myc*, and *E2F*, which we describe in a new supplemental figure, S10 (in fact, we had performed several of these experiments previously but due to the reasons described below had chosen not to include in our initial submission.)

Unfortunately, these experiments were not informative for our study because none of these manipulations effectively blocked endocycling nearly as strongly as *InR-RNAi*. *fzr-RNAi* (using

the same line used in e.g. PMID: 25142462) did not show any decrease in ploidy after 14 days of RNAi at 29°C, indicating that this gene is not strictly required for endocycling in the proventriculus. *E2F1* RNAi did cause a decrease in ploidy, but not as dramatically as *InR-i* and therefore not particularly informative. Myc plays many different roles in *Drosophila*, and we found that while *myc* RNAi did lead to dramatic loss of EdU+ cells, it also caused a severe morphological phenotype in which the tissue structure of Zone 4 itself was highly disrupted, with GFP+ cells nearly abolished. Thus, *myc* RNAi was not an effective means to cleanly block endocycling.

For *cycE* RNAi, we tested three independent RNAi lines (BL38902, BL33654, and BL29314). Two of these lines (BL38902 and BL33654) had no consistent effect on EdU+ cells or ploidy, and we do not include in the manuscript. The third, BL29314, produced a more complex phenotype than simply reducing endocycling, making these difficult to interpret in the context of our study. Specifically, while there was a reduction in endocycling and the overall tissue size of Zone 4, these effects were muted compared to *InR* RNAi, and we also observed large aggregates of peritrophic membrane components throughout Zone 4 (shown in Figure S10) which stained positive for all three markers of PM. While these results may be interesting in their own right, they do not allow us to cleanly assay the effect of blocking endocycling using *cycE* RNAi.

It is possible that there are technical explanations for all these results, such as perhaps that RNAi knockdowns of these factors are not sufficiently strong over 14-21d time period. However, the ultimate result is that at this point, the only genetic manipulation that we have found which consistently blocks endocycling in Zone 4 cells is *InR* RNAi. We have also added text in the manuscript to clarify that *InR* may play other roles in these cells in addition to solely blocking endocycling.

6. The authors could use the dextran permeability assay (Kuraishi et al. PNAS 2011) to determine if the peritrophic membrane is compromised when endoreplication is upregulated or inhibited. It is unclear if the changes in lectin staining are indicative of a functional change.

We have now performed both the Smurf assay and a fluorescent bead assay similar to that performed by Kuraishi *et al* (2011), and both assays did not reveal a significant disruption to the gut barrier in flies with endocycling inhibited. Together with our finding that these flies are more susceptible to pathogenic infection, we conclude that endocycling (and the resultant increase in synthetic capacity) is not strictly required for the synthesis of a functional peritrophic membrane under standard conditions, but instead provides robustness in the face of environmental challenges that flies may encounter including infection.

Minor Comments:

1. The authors should consider moving Figure 6 (GTRACE) to the supplement as it is redundant with Figure 4 (MARCM).

Done.

2. Line 333-335: Liver hepatocytes are derived from the endoderm and well-documented to endoreplicate in response to tissue damage, so there is insufficient evidence to make a claim of an ectoderm vs endoderm difference in endoreplication.

We have removed this speculative statement from the manuscript.

3. Miss cited figures (i.e. Fig S6C – line 397-8 does not exist). Figure S7 legend is missing a description for part (C).

Fixed.

4. Several graphs (i.e. Figure 1H and 6F) are missing significance and the statistical test if used it is not reported.

Fixed.

5. Check image stains are labeled in all figures (i.e. Figure 2E is missing this info)

Fixed.

6. It would be helpful to provide the guts (n) and PH3+ nuclei (#) for intestines in Figure 3B as well.

Done - under these conditions, >600 ph3+ ISC's are observed in midguts.

Reviewer #4 (Remarks to the Author):

In this thorough and carefully conducted study, the authors investigate the mechanism by which tissue homeostasis is maintained in the *Drosophila* adult proventriculus. A previous study reported on the presence of BrdU+ cells in the proventriculus and proposed that it therefore contains mitotically active stem cells. Here the authors reexamine this conclusion and provide convincing evidence that these cells are endocycling rather than undergoing mitosis and therefore are not stem cells. They reach this conclusion through a combination of approaches, including stains for mitotic markers and clonal analysis in a wide range of experimental conditions designed to enrich for mitotic cells if they exist in this tissue. However, despite this extensive search, they do not find evidence of cells in mitosis. Suitable positive controls are included to confirm that their methods are working as expected. They also generate a single cell atlas of the proventriculus and use the results, along with a screen and a recently developed split-Gal4 tool to specifically target the region of the tissue where cells in S-phase can be found. Interestingly, they find that cells in this region are polyploid and that the ploidy increases with age, suggesting that they continue to endocycle during adulthood. The authors propose that this may be a mechanism to compensate for age-dependent cell loss, which they provide evidence for. Overall, I think this study will be a valuable contribution to the field and I am supportive of publication. I have only a few minor comments, which I recommend the authors consider before publication:

We appreciate the Reviewer's supportive feedback and helpful suggestions.

1. I am confused by the results described in the paragraph from lines 132-139 (Figure 1D is referenced but I think the authors are referring to Figure 1F): if the cells are continuously endocycling, why would the size of EdU+ nuclei increase over 5 days? Wouldn't there be a distribution of nuclear sizes at any given time point when EdU is administered such that both large and small nuclei could be labeled even at day 1?

We appreciate this helpful feedback, and as all Reviewers pointed out flaws with the analyses of these panels we have removed them from the revised manuscript.

2. On line 147, it seems that Figure 1H should be referenced (not Figure 1E).

Fixed.

3. On Line 252-253, the authors state that they never saw cells travel from Zone 4 to any other portion of the proventriculus but, without live imaging or clonal analysis, how is it possible to track cell movement?

We thank the Reviewer for raising this issue. We have clarified in the text that this conclusion is based on the fact that we never observed GFP+ cells located outside of Zone 4 despite increased cell cycling and tissue growth. Had these cells migrated outside of Zone 4 as claimed by Singh et al 2011, we would have expected to see GFP+ cells located elsewhere in the proventriculus.

4. The data in Fig 4D show that there are, in fact, a small number of large clones in the proventriculus at 14 days after heat shock (dark grey bar at the top of the third column from the left) but this is not discussed in the text. How did these larger clones arise if the cells cannot undergo mitosis?

We have added text in the manuscript to clarify that we believe this single example of large clone was the result of a spontaneous clone formation earlier in development.

5. The code used to analyze the scRNAseq data as well as the final Seurat objects should be made available.

We have now made this code available and provided a link to the scripts in the manuscript.

Sincerely,

Ben Ewen-Campen & Norbert Perrimon

Author response to reviewers, following resubmission: “The *Drosophila* proventriculus lacks stem cells but compensates for age-related cell loss via endoreplication-mediated cell growth”

REVIEWERS' COMMENTS

Reviewer #1 (Remarks to the Author):

The changes made in the revised version have addressed the major concerns raised in the original review in a satisfactory manner. I now favor publication without further substantive changes.

The paper is framed as a re-investigation of a 2011 report describing proventriculus (PV) stem cells that were never confirmed and fell into obscurity. However, this paper goes beyond classical stem cells and provides convincing evidence of a different and very interesting alternative phenomenon related to "wound-induced polyploidy," an alternative repair system based on induced endocycles rather than mitotic cycles. Midgut and hindgut enterocytes, and many other cells are known under some conditions to polyploidize in response to tissue damage, as has now been well studied previously by multiple groups including the authors. Here they show that PV4 cells undergo scattered endocycling in a ring-like pattern during much of adult life, in the absence of a known source of tissue damage.

An interesting potential explanation would be that active peritrophic membrane (PM) production is intrinsically damaging to the posterior rim of PV4 cells. The cells may be responding to cell loss with induced polyploidization of some remaining nearby PV4 cells to maintain the functional capacity of this region, which is thought to play a critical role by generating the major chitin-rich inner layer of the peritrophic membrane. This layer is involved in digestion, and restricting access of gut microbiota to enterocyte tissue. These findings recall the well known theme in studies of the gut that the normal tasks of some cells are intrinsically damaging. This is the generally accepted reason for the rapid turnover of midgut enterocytes and the presence of highly active stem cells to supply replacements. Damaging food and microorganisms are thought to be largely excluded from the upper PV, and this is a rationale for the absence of active stem cells. Something must be different in this region of PV4, and the capacity of the endocycle-based repair may eventually be inadequate and contribute to an age and metabolism-related loss of homeostasis and normal tissue structure. A similar scenario of polyploidization incompletely counteracting damage leading to slow progression of disease was described in the mouse model of Fuchs Endothelial Corneal Dystrophy analyzed in reference 14. Endocycling in the pattern described might serve as a marker for such early aging.

The authors in the revised version make a strong case for cell turnover and compensatory

polyploidization (rather than cell generation) in PV4. They now show that the average ploidy as well as nuclear volume of zone 4 cells increases over the lifespan. They do not currently propose a reason for this behavior other than to call it a "unique pattern of endoreplication." I think the paper would be further improved by a little more discussion of what factors might lead to such a unique pattern of cell cycle behavior.

Author response: We are grateful for these positive comments and have now included additional discussion based on the suggestion of the reviewer's interesting proposal that PM production may be intrinsically damaging.

Reviewer #3 (Remarks to the Author):

Summary:

This revised manuscript addresses the major and minor concerns of the original study. The ploidy analysis now proves that zone 4 cells in the proventriculus endocycle with age and appear to do so to compensate for cell loss. The authors find that cell loss occurs naturally with age as well as in response to genetically induced-apoptosis. This is consistent with other studies performed to date in fruit fly. This study also provides a comprehensive analysis of the proventriculus cellular organization, its gene expression program, as well as identification of a novel Gal4 driver to regulate gene expression in zone 4. The study debunks previous report of stem cells in the tissue and finds that tissue homeostasis appears to be maintained by cell growth, instead of cell division. Despite the many different genetic manipulations performed, the authors were not able to find a condition to directly inhibit endoreplication in zone 4 cells. Therefore, the physiological importance of endoreplication to the intestine's peritrophic membrane synthesis as well as infection resistance remain inconclusive lessening the impact of this study.

Remaining Comments:

1. A major conclusion of this paper is that endoreplication with age or injury compensates for cell loss. The authors should be able to definitively show this by summing the nuclear ploidy in young vs old or uninjured vs repaired tissues. If ploidy is compensating for cell loss, then the total ploidy of the two conditions will remain unchanged despite a reduction in cell number.
2. The authors still claim that their findings are unique (line 625-635) from other studies on injury or age-induced polyploidy. However, the uniqueness is subtle at best as Nandakumar et al. 2020 demonstrated that neuron ploidy increases with age via endocycle. As for nutrient dependence, endoreplication is well-characterized to be sensitive to nutritional inputs. Therefore, the changes observed are more likely tissue intrinsic and due to cell type specific differences in

longevity/ turnover rate.

Author response:

Re (1): We are grateful for these additional comments. While we would very much wish to calculate the total ploidy of Zone 4 during aging and damage, unfortunately our experimental setup does not allow this calculation. Specifically, due to the 3D shape of the tissue, we are unable to comprehensively measure the ploidy of Zone 4 because multiple nuclei overlap with each other in 2D and 3D, and with Zone 3 and the esophagus. Instead, we have relied on measuring the ploidy of as many nuclei as can be unambiguously scored in each sample.

Re (2): We have modified our discussion in line with the reviewer's context including Nandakumar.

Reviewer #4 (Remarks to the Author):

With this revision, the authors have fully addressed my concerns.

We appreciate the support of Reviewer #4.